# Virtual Brain Inference (VBI), a flexible and integrative toolkit for efficient probabilistic inference on whole-brain models

Abolfazl Ziaeemehr[1]*, Marmaduke Woodman[1], Lia Domide[2], Spase Petkoski[1], Viktor Jirsa[1]*[†], Meysam Hashemi[1]*[†]

[1]Aix Marseille University, INSERM, INS, Inst Neurosci System, Marseille, France; [2]Codemart, Cluj-Napoca, Romania

*For correspondence:
abolfazl.ziaee-mehr@univ-amu.
fr (AZ);
viktor.jirsa@univ-amu.fr (VJ);
meysam.hashemi@univ-amu.fr
(MH)

†These authors contributed
equally to this work

Competing interest: The authors
declare that no competing
interests exist.

Reviewing Editor: Alex Fornito,
Monash University, Melbourne,
Australia

## eLife Assessment

This paper presents a **valuable** software package, named "Virtual Brain Inference" (VBI), that enables faster and more efficient inference of parameters in dynamical system models of whole-brain activity, grounded in artificial network networks for Bayesian statistical inference. The authors have provided **convincing** evidence, across several case studies, for the utility and validity of the methods using simulated data from several commonly used models, but more thorough benchmarking could be used to demonstrate the practical utility of the toolkit. This work will be of interest to computational neuroscientists interested in modelling large-scale brain dynamics.

**Abstract** Network neuroscience has proven essential for understanding the principles and mechanisms underlying complex brain (dys)function and cognition. In this context, whole-brain network modeling—also known as virtual brain modeling—combines computational models of brain dynamics (placed at each network node) with individual brain imaging data (to coordinate and connect the nodes), advancing our understanding of the complex dynamics of the brain and its neurobiological underpinnings. However, there remains a critical need for automated model inversion tools to estimate control (bifurcation) parameters at large scales associated with neuroimaging modalities, given their varying spatio-temporal resolutions. This study aims to address this gap by introducing a flexible and integrative toolkit for efficient Bayesian inference on virtual brain models, called Virtual Brain Inference (VBI). This open-source toolkit provides fast simulations, taxonomy of feature extraction, efficient data storage and loading, and probabilistic machine learning algorithms, enabling biophysically interpretable inference from non-invasive and invasive recordings. Through in-silico testing, we demonstrate the accuracy and reliability of inference for commonly used whole-brain network models and their associated neuroimaging data. VBI shows potential to improve hypothesis evaluation in network neuroscience through uncertainty quantification and contribute to advances in precision medicine by enhancing the predictive power of virtual brain models.

## Introduction

Understanding the complex dynamics of the brain and their neurobiological underpinnings, with the potential to advance precision medicine (*Falcon et al., 2016*; *Tan et al., 2016*; *Vogel et al., 2023*; *Williams and Whitfield Gabrieli, 2025*), is a central goal in neuroscience. Modeling these dynamics provides crucial insights into causality and mechanisms underlying both normal brain function and

various neurological disorders (*Breakspear, 2017*; *Wang et al., 2023b*; *Ross and Bassett, 2024*). By integrating the average activity of large populations of neurons (e.g. neural mass models; *Wilson and Cowan, 1972*; *Jirsa and Haken, 1996*; *Deco et al., 2008*; *Jirsa et al., 2014*; *Montbrió et al., 2015*; *Cook et al., 2022*) with information provided by structural imaging modalities (i.e. connectome; *Honey et al., 2009*; *Sporns et al., 2005*; *Schirner et al., 2015*; *Bazinet et al., 2023*), the whole-brain network modeling has proven to be a powerful tractable approach for simulating brain activities and emergent dynamics as recorded by functional imaging modalities (such as (s)EEG, MEG, and fMRI; *Sanz-Leon et al., 2015*; *Schirner et al., 2022*; *Amunts et al., 2022*; *D'Angelo and Jirsa, 2022*; *Patow et al., 2024*; *Hashemi et al., 2025*).

The whole-brain models have been well-established in network neuroscience (*Sporns, 2016*; *Bassett and Sporns, 2017*) for understanding the brain structure and function (*Ghosh et al., 2008*; *Honey et al., 2010*; *Park and Friston, 2013*; *Melozzi et al., 2019*; *Suárez et al., 2020*; *Feng et al., 2024*; *Tanner et al., 2024*) and investigating the mechanisms underlying brain dynamics at rest (*Deco et al., 2011*; *Wang et al., 2019*; *Ziaeemehr et al., 2020*; *Kong et al., 2021*), normal aging (*Lavanga et al., 2023*; *Zhang et al., 2024*), and also altered states such as anesthesia and loss of consciousness (*Barttfeld et al., 2015*; *Hashemi et al., 2017*; *Luppi et al., 2023*; *Perl et al., 2023b*). This class of computational models, also known as virtual brain models (*Jirsa et al., 2010*; *Sanz Leon et al., 2013*; *Sanz-Leon et al., 2015*; *Schirner et al., 2022*; *Jirsa et al., 2023*; *Wang et al., 2024*), has shown remarkable capability in delineating the pathophysiological causes of a wide range of brain diseases, such as epilepsy (*Jirsa et al., 2017*; *Proix et al., 2017*; *Wang et al., 2023b*), multiple sclerosis (*Wang et al., 2024*; *Mazzara et al., 2025*), Alzheimer's disease (*Yalçınkaya et al., 2023*; *Perl et al., 2023a*), Parkinson's disease (*Jung et al., 2022*; *Angiolelli et al., 2025*), neuropsychiatric disorders (*Deco and Kringelbach, 2014*; *Iravani et al., 2021*), stroke (*Allegra Mascaro et al., 2020*; *Idesis et al., 2022*), and focal lesions (*Rabuffo et al., 2025*). In particular, they enable the personalized simulation of both normal and abnormal brain activities, along with their associated imaging recordings, thereby stratifying between healthy and diseased states (*Liu et al., 2016*; *Patow et al., 2023*; *Perl et al., 2023a*) and potentially informing targeted interventions and treatment strategies (*Jirsa et al., 2017*; *Proix et al., 2018*; *Wang et al., 2023b*; *Jirsa et al., 2023*; *Hashemi et al., 2025*). Although there are only a few tools available for forward simulations at the whole-brain level, for example the brain network simulator The Virtual Brain (TVB; *Sanz Leon et al., 2013*), there is a lack of tools for addressing the inverse problem, that is finding the set of control (generative) parameters that best explains the observed data. This study aims to bridge this gap by addressing the inverse problem in large-scale brain networks, a crucial step toward making these models operable for clinical applications.

Accurately and reliably estimating the parameters of whole-brain models remains a formidable challenge, mainly due to the high dimensionality and nonlinearity inherent in brain activity data, as well as the non-trivial effects of noise and network inputs. A large number of previous studies in whole-brain modeling have relied on optimization techniques to identify a single optimal value from an objective function, scoring the model's performance against observed data (*Wang et al., 2019*; *Kong et al., 2021*; *Cabral et al., 2022*; *Liu et al., 2023*). This approach often involves minimizing metrics such as the Kolmogorov-Smirnov distance or maximizing the Pearson correlation between observed and generated data features such as functional connectivity (FC), functional connectivity dynamics (FCD), and/or power spectral density (PSD). Although fast, such a parametric approach results in only point estimates and fails to capture the relationship between parameters and their associated uncertainty. This limits the generalizability of findings and hinders identifiability analysis, which explores the uniqueness of solutions. Furthermore, optimization algorithms can easily get stuck in local extrema, requiring multi-start strategies to address potential parameter degeneracies. These additional steps, while necessary, ultimately increase the computational cost. Critically, the estimation heavily depends on the form of the objective function defined for optimization (*Svensson et al., 2012*; *Hashemi et al., 2018*). These limitations can be overcome by employing Bayesian inference, which naturally quantifies the uncertainty in the estimation and statistical dependencies between parameters, leading to more robust and generalizable models. Bayesian inference is a principal method for updating prior beliefs with information provided by data through the likelihood function, resulting in a posterior probability distribution that encodes all the information necessary for inferences and predictions. This approach has proven essential for understanding the intricate relationships between brain structure and function (*Hashemi et al., 2021*; *Lavanga et al., 2023*; *Rabuffo et al., 2025*), as well as for revealing the

pathophysiological causes underlying brain disorders (*Hashemi et al., 2023*; *Yalçınkaya et al., 2023*; *Wang et al., 2024*; *Wang et al., 2024*; *Hashemi et al., 2025*; *Hashemi et al., 2024*).

In this context, simulation-based inference (SBI; *Cranmer et al., 2020*; *Gonçalves et al., 2020*; *Hashemi et al., 2023*; *Hashemi et al., 2024*) has gained prominence as an efficient methodology for conducting Bayesian inference in complex models where traditional inference techniques become inapplicable. SBI leverages computational simulations to generate synthetic data and employs advanced probabilistic machine learning methods to infer the joint distribution over parameters that best explain the observed data, along with associated uncertainty. This approach is particularly well-suited for Bayesian inference on whole-brain models, which often exhibit complex dynamics that are difficult to retrieve from neuroimaging data with conventional estimation techniques. Crucially, SBI circumvents the need for explicit likelihood evaluation and the Markovian (sequential) property required in sampling. Markov chain Monte Carlo (MCMC; *Gelman et al., 1995*) is the gold-standard nonparametric technique and asymptotically exact for sampling from a probability distribution. However, for Bayesian inference on whole-brain models given high-dimensional data, the likelihood function becomes intractable, rendering MCMC sampling computationally prohibitive. SBI offers significant advantages, such as parallel simulation while leveraging amortized learning, making it effective for personalized inference from large datasets (*Hashemi et al., 2024*). Amortization in artificial neural networks refers to the idea of reusing learned computations across multiple tasks or inputs (*Gershman and Goodman, 2014*). Amortization in Bayesian inference refers to the process of training a shared inference network (e.g. a neural network) with an intensive upfront computational cost, to perform fast inference across many different observations. Instead of re-running inference for each new observation, the trained model can rapidly return posterior estimates, significantly reducing computational cost at test time. Following an initial computational cost during simulation and training to learn all posterior distributions, subsequent evaluation of new hypotheses can be conducted efficiently, without additional computational overhead for further simulations (*Hashemi et al., 2023*). Importantly, SBI sidesteps the convergence issues caused by complex geometries that are often encountered when using gradient-based MCMC methods (*Betancourt and Girolami, 2013*; *Betancourt et al., 2014*; *Hashemi et al., 2020*). It also substantially outperforms approximate Bayesian computation (ABC) methods, which rely on a threshold to accept or reject samples (*Sisson et al., 2007*; *Beaumont et al., 2009*; *Gonçalves et al., 2020*). Such a likelihood-free approach provides us with generic inference on complex systems as long as we can provide three modules:

1. A prior distribution, describing the possible range of parameters from which random samples can be easily drawn, that is $\vec{\theta} \sim p(\vec{\theta})$.
2. A simulator in computer code that takes parameters as input and generates data as output, that is $\vec{x} \sim p(\vec{x} \mid \vec{\theta})$.
3. A set of low-dimensional data features, which are informative of the parameters that we aim to infer.

These elements prepare us with a training data set $\{(\vec{\theta}_i, \vec{x}_i)\}_{i=1}^{N_{sim}}$ with a budget of $N_{sim}$ simulations. Then, using a class of *deep neural density estimators*, such as masked autoregressive flows (MAFs; *Papamakarios and Pavlakou, 2017*) or neural spline flows (NSFs; *Durkan et al., 2019*), we can approximate the posterior distribution of parameters given a set of observed data, that is $p(\vec{\theta} \mid \vec{x}_{obs})$. Therefore, a versatile toolkit should be flexible and integrative, adeptly incorporating these modules to enable efficient Bayesian inference over complex models.

To address the need for widely applicable, reliable, and efficient parameter estimation from different (source-localized) neuroimaging modalities, we introduce Virtual Brain Inference (VBI), a flexible and integrative toolkit for probabilistic inference at whole-brain level. This open-source toolkit offers fast simulation through just-in-time (JIT) compilation of various brain models in different programming languages (Python/C++) and devices (CPUs/GPUs). It supports space-efficient storage of simulated data (HDF5/NPZ/PT), provides a memory-efficient loader for batched data, and facilitates the extraction of low-dimensional data features (FC/FCD/PSD). Additionally, it enables the training of deep neural density estimators (MAFs/NSFs), making it a versatile tool for inference on neural sources corresponding to (s)EEG, MEG, and fMRI recordings. VBI leverages high-performance computing, significantly enhancing computational efficiency through parallel processing of large-scale datasets, which would be impractical with current alternative methods. Although SBI has been used for low-dimensional parameter spaces (*Gonçalves et al., 2020*; *Wang et al., 2024*; *Baldy et al., 2024*), we

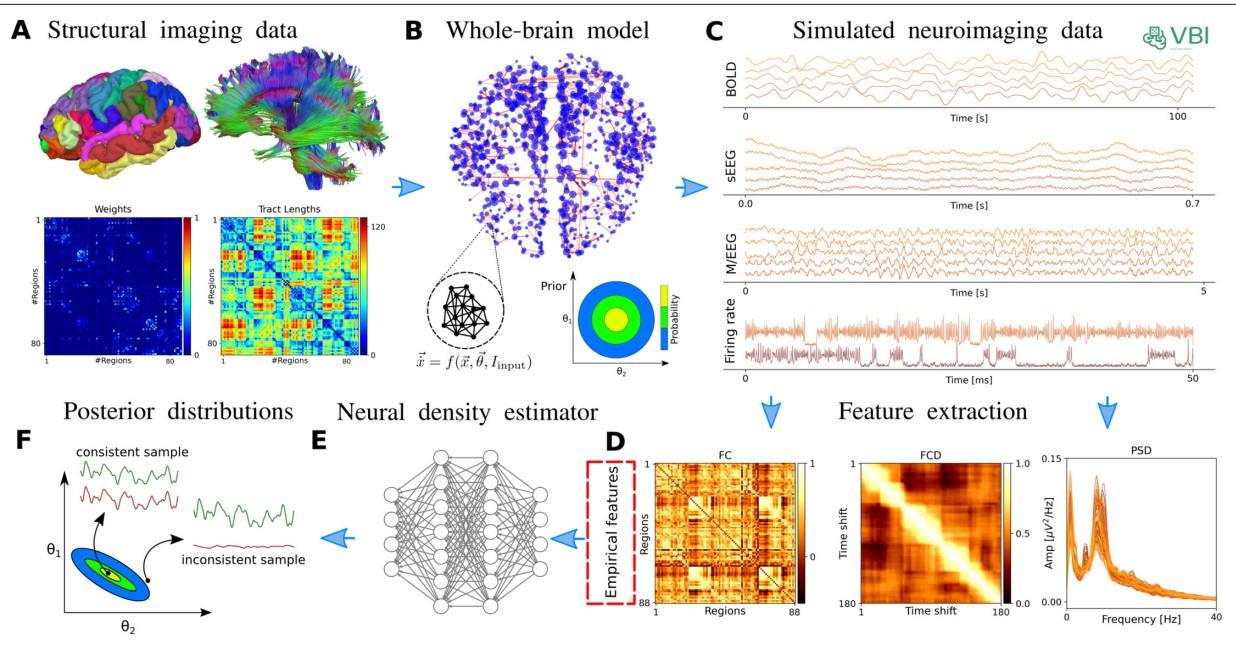

**Figure 1.** The workflow of Virtual Brain Inference (VBI). This probabilistic approach is designed to estimate the posterior distribution of control parameters in virtual brain models from whole-brain recordings. (**A**) The process begins with constructing a personalized connectome using diffusion tensor imaging and a brain parcellation atlas, such as Desikan-Killiany (**Desikan et al., 2006**), Automated Anatomical Labeling (**Tzourio-Mazoyer et al., 2002**), or VEP (**Wang et al., 2021**). (**B**) The personalized virtual brain model is then assembled. Neural mass models describing the averaged activity of neural populations, in the generic form of $\vec{x} = f(\vec{x}, \vec{\theta}, I_{input})$, are placed to each brain region and interconnected via the structural connectivity matrix. Initially, the control parameters are randomly drawn from a simple prior distribution. (**C**) Next, the VBI operates as a simulator that uses these samples to generate time series data associated with neuroimaging recordings. (**D**) We extract a set of summary statistics from the low-dimensional features of the simulations (FC, FCD, PSD) for training. (**E**) Subsequently, a class of deep neural density estimators is trained on pairs of random parameters and their corresponding data features to learn the joint posterior distribution of the model parameters. (**F**) Finally, the amortized network allows us to quickly approximate the posterior distribution for new (empirical) data features, enabling us to make probabilistic predictions that are consistent with the observed data.

demonstrate that it can scale to whole-brain models with high-dimensional unknown parameters, as long as informative data features are provided. VBI is now accessible on the cloud platform EBRAINS (https://ebrains.eu), enabling users to explore more realistic brain dynamics underlying brain (dys) functioning using Bayesian inference.

In the following sections, we will describe the architecture and workflow of the VBI toolkit and demonstrate the validation through a series of case studies using in silico data. We explore various whole-brain models corresponding to different types of brain recordings: a whole-brain network model of Wilson-Cowan (**Wilson and Cowan, 1972**), Jansen-Rit (**Jansen and Rit, 1995**; **David and Friston, 2003**), and Stuart-Landau (**Selivanov et al., 2012**) for simulating neural activity associated with EEG/MEG signals, the Epileptor (**Jirsa et al., 2014**) related to stereoelectro-EEG (sEEG) recordings, and Montbrió (**Montbrió et al., 2015**), and Wong-Wang (**Wong and Wang, 2006**; **Deco et al., 2013**) mapped to fMRI BOLD signals. Although these models represent source signals and could be applied to other modalities (e.g. Stuart-Landau representing generic oscillatory dynamics), we focused on their capabilities to perform optimally in specific contexts. For instance, some are better suited for encephalographic signals (e.g. EEG/MEG) due to their ability to preserve spectral properties, while others have been used for fMRI data, emphasizing their ability to capture dynamic features such as bistability and time-varying functional connectivity.

## VBI workflow

*Figure 1* illustrates an overview of our approach in VBI, which combines virtual brain models and SBI to make probabilistic predictions on brain dynamics from (sourc-localized) neuroimaging recordings. The inputs to the pipeline include the structural imaging data (for building the connectome), functional imaging data such as (s)EEG/MEG, and fMRI as the target for fitting, and prior information as a

plausible range over control parameters for generating random simulations. The main computational costs involve model simulations and data feature extraction. The output of the pipeline is the joint posterior distribution of control parameters (such as excitability, synaptic weights, or effective external input) that best explains the observed data. Since the approach is amortized (i.e. it learns across all combinations in the parameter space), it can be readily applied to any new data from a specific subject.

In the first step, non-invasive brain imaging data, such as T1-weighted MRI and Diffusion-weighted MRI (DW-MRI), are collected for a specific subject (**Figure 1A**). T1-weighted MRI images are processed to obtain brain parcellation, while DW-MRI images are used for tractography. Using the estimated fiber tracts and the defined brain regions from the parcellation, the connectome (i.e. the complete set of links between brain regions) is constructed by counting the fibers connecting all regions. The SC matrix, with entries representing the connection strength between brain regions, forms the structural component of the virtual brain which constrains the generation of brain dynamics and functional data at arbitrary brain locations (e.g. cortical and subcortical structures).

Subsequently, each brain network node is equipped with a computational model of average neuronal activity, known as neural mass models (see **Figure 1B** and Materials and methods). They can be represented in the generic form of a dynamical model as $\dot{\vec{x}} = f(\vec{x}, \vec{\theta}, I_{input})$, with the system variables $\vec{x}$ (such as membrane potential and firing rate), the control parameters $\vec{\theta}$ (such as excitability), and the input current $I_{input}$ (such as stimulation). This integration of mathematical mean-field modeling (neural mass models) with anatomical information (connectome) allows us to efficiently analyze functional neuroimaging modalities at the whole-brain level.

To quantify the posterior distribution of control parameters given a set of observations, $p(\vec{\theta} \mid \vec{x})$, we first need to define a plausible range for the control parameters based on background knowledge $p(\vec{\theta})$, that is a simple base distribution known as a prior. We draw random samples from the prior and provide them as input to the VBI simulator (implemented by *Simulation* module) to generate simulated time series associated with neuroimaging recordings, as shown in **Figure 1C**. Subsequently, we extract low-dimensional data features (implemented by *Features* module), as shown in **Figure 1D** for FC/FCD/PSD, to prepare the training dataset $\{(\vec{\theta}_i, \vec{x}_i)\}_{i=1}^{N_{sim}}$, with a budget of $N_{sim}$ simulations. Then, we use a class of deep neural density estimators, such as MAF or NSF models, as schematically shown in **Figure 1E**, to learn all the posterior $p(\vec{\theta} \mid \vec{x})$. Finally, we can readily sample from $p(\vec{\theta} \mid \vec{x}_{obs})$, which determines the probability distribution in parameter space that best explains the observed data.

**Figure 2** depicts the structure of the VBI toolkit, which consists of three main modules. The first module, referred to as the *Simulation* module, is designed for fast simulation of whole-brain models, such as Wilson-Cowan (Wilson-Cowan model), Jansen-Rit (Jansen-Rit model), Stuart-Landau

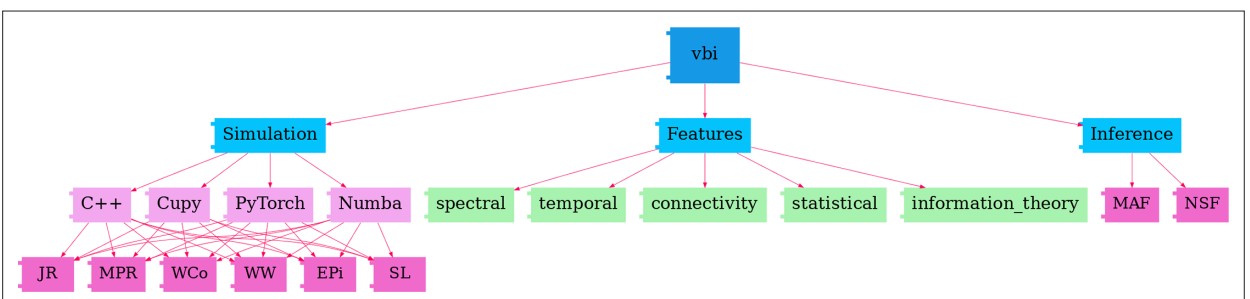

**Figure 2.** Flowchart of the VBI Structure. This toolkit consists of three main modules: (1) The *Simulation* module, implementing various whole-brain models, such as Wilson-Cowan (WCo), Jansen-Rit (JR), Stuart-Landau (SL), Epileptor (EPi), Montbrió (MPR), and Wong-Wang (WW), across different numerical computing libraries (C++, Cupy, PyTorch, Numba). (2) The *Features* module, offering an extensive toolbox for extracting low-dimensional data features, such as spectral, temporal, connectivity, statistical, and information theory features. (3) The *Inference* module, providing neural density estimators (such as MAF and NSF) to approximate the posterior of parameters.

The online version of this article includes the following figure supplement(s) for figure 2:

**Figure supplement 1.** Benchmarking the computational cost of CPU/GPU and MAF/NSF.

**Figure supplement 2.** Evaluating the accuracy of estimation (parameter $G$) with respect to the number of simulations used for training the MAF/NSF density estimator by (**A**) posterior shrinkage and (**B**) posterior z-score, versus the number of simulations.

**Figure supplement 3.** Estimation of the global coupling parameter across whole-brain models.

(Stuart-Landau oscillator), Epileptor (Epileptor model), Montbrió (Montbrió model), and Wong-Wang (Wong-Wang model). These whole-brain models are implemented across various numerical computing libraries such as Cupy (GPU-accelerated computing with Python), C++ (a high-performance systems programming language), Numba (a JIT compiler for accelerating Python code), and PyTorch (an open-source machine learning library for creating deep neural networks).

The second module, *Features*, provides a versatile tool for extracting low-dimensional features from simulated time series (see Comprehensive feature extraction). The features include, but are not limited to, *spectral*, *temporal*, *connectivity*, *statistical*, and *information theory* related features, and the associated summary statistics. The third module focuses on *Inference*, that is training the deep neural density estimators, such as MAF and NSF (see Simulation-based inference), to learn the joint posterior distribution of control parameters. See *Figure 2—figure supplement 1* and *Figure 2—figure supplement 2* for benchmarks comparing CPU/GPU and MAF/NSF performances, and *Figure 2—figure supplement 3* for the estimation of the global coupling parameter across different whole-brain network models, evaluated under multiple configurations.

## Results

In the following, we demonstrate the capability of VBI for inference on the state-of-the-art whole-brain network models using in silico testing, where the ground truth is known. We apply this approach to simulate neural activity and associated measurements, including (s)EEG/MEG and fMRI, while also providing diagnostics for the accuracy and reliability of the estimation. Note that for (s)EEG/MEG neuroimaging, we perform inference at the regional level rather than at the sensor level, whereas for fMRI, it is mapped using the Balloon-Windkessel model (see The Balloon-Windkessel model). The results presented are based on synthetic data generated using a set of predefined parameters, referred to as the ground truth, randomly selected within biologically plausible ranges and incorporating a certain level of heterogeneity.

### Whole-brain network of Wilson-Cowan model

We first demonstrate inference on the whole-brain network model of the Wilson-Cowan (see *Equation 1*), which is capable of generating a wide range of oscillatory dynamics depending on the control parameters. Specifically, we estimate the bifurcation parameters $P_i \in \mathbb{R}^{88}$, representing the external input to each excitatory population, and the global excitatory coupling parameter $g_e$. *Figure 3A and B* present the observed and predicted EEG-like signals, represented by the activity of excitatory populations $E$ across regions, and *Figure 3C and D* show the corresponding power spectral density (PSD), as data features. *Figure 3E and F* illustrate the inferred posterior distributions for parameters $P_i$ and $g_e$, respectively, given $\theta = \{g_e, P_i\} \in \mathbb{R}^{89}$. For training, we conducted 250 k random simulations from uniform priors $g_e \sim \mathcal{U}(0, 3)$ and $P_i \sim \mathcal{U}(0, 3)$. After approximately 2 hr of training using MAF density estimators, posterior sampling was completed within a few seconds. Due to the large number of simulations and informativeness of data features, we achieved accurate estimations of the high-dimensional and heterogeneous control parameters. Ground-truth values (shown in green) are well recovered, leading to close agreement between observed and predicted PSDs of the signals. Finally, *Figure 3G* reports the posterior shrinkage and z-score metrics used to evaluate the quality of the parameter estimation. The results indicate that the inferred posteriors are both precise and well-centered around the ground-truth values, as reflected by high shrinkage and low z-scores. See *Figure 3—figure supplement 1* for estimation over other configurations. Moreover, *Figure 3—figure supplement 2*, and *Figure 3—figure supplement 3*, show the estimations by ignoring the spatial information in the data features, indicating the higher accuracy of NSF, though with substantially more computational cost for training compared to MAF.

### Whole-brain network of Jansen-Rit model

Then, we demonstrate the inference on heterogeneous control parameters in the whole-brain network of Jansen-Rit (see *Equation 2*), commonly used for modeling EEG/MEG data, for example in dementia and Alzheimer's disease (*Triebkorn et al., 2022*; *Stefanovski et al., 2019*). *Figure 4A and B* show the observed and predicted EEG signals, given by $y_{1i} - y_{2i}$ at each region, while *Figure 4C and D* illustrate the observed and predicted features such as PSD, respectively. *Figure 4E and F* show the estimated

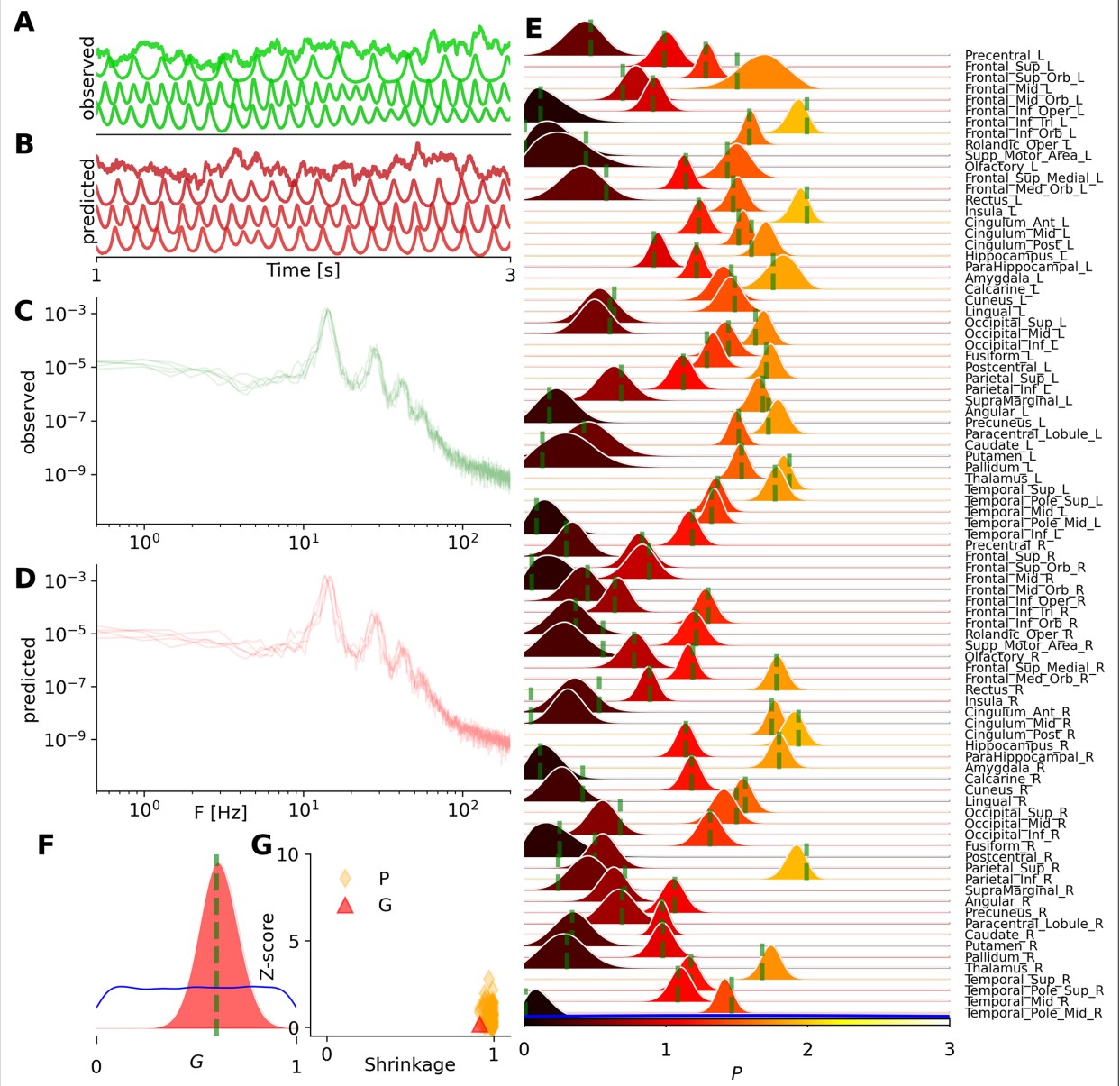

**Figure 3.** Bayesian inference on heterogeneous control parameters in the whole-brain network of Wilson-Cowan model. The set of inferred parameters is $\vec{\theta} = \{G, P_i\} \in \mathbb{R}^{89}$, with the global scaling parameter $G$ and average external input current to excitatory populations $P_i$ per region, given $i \in \{1, 2, ..., N_n = 88\}$ parcelled regions. Summary statistics of power spectrum density (PSD) were used for training of MAF density estimators, with a budget of 260 k simulations. (**A**) and (**B**) illustrate the observed and predicted neural activities, respectively. (**C**) and (**D**) show the observed and predicted PSDs as the data features. (**E**) and (**F**) display the posterior distribution of $P_i$ per region, and global coupling $G$, respectively. The ground truth and prior are represented by a vertical green line and a blue distribution, respectively. (**G**) shows the inference evaluation using posterior shrinkage and z-score.

The online version of this article includes the following figure supplement(s) for figure 3:

**Figure supplement 1.** Inference on heterogeneous control parameters across different configurations.

**Figure supplement 2.** Inference using the MAF estimator, but ignoring the spatial information in the data features.

**Figure supplement 3.** Inference using the NSF estimator, but ignoring the spatial information in the data features.

posterior distributions of synaptic connections $C_i$, and the global coupling parameter $G$, respectively, given the set of unknown parameters $\vec{\theta} = \{G, C_i\} \in \mathbb{R}^{89}$. Here we conducted 50 k random simulations with samples drawn from uniform priors $G \in \mathcal{U}(0, 5)$ and $C_i \in \mathcal{U}(100, 650)$. After approximately 45 min of training (MAF density estimator), the posterior sampling took only a few seconds. With such a sufficient number of simulations and informative data features, VBI shows accurate estimation

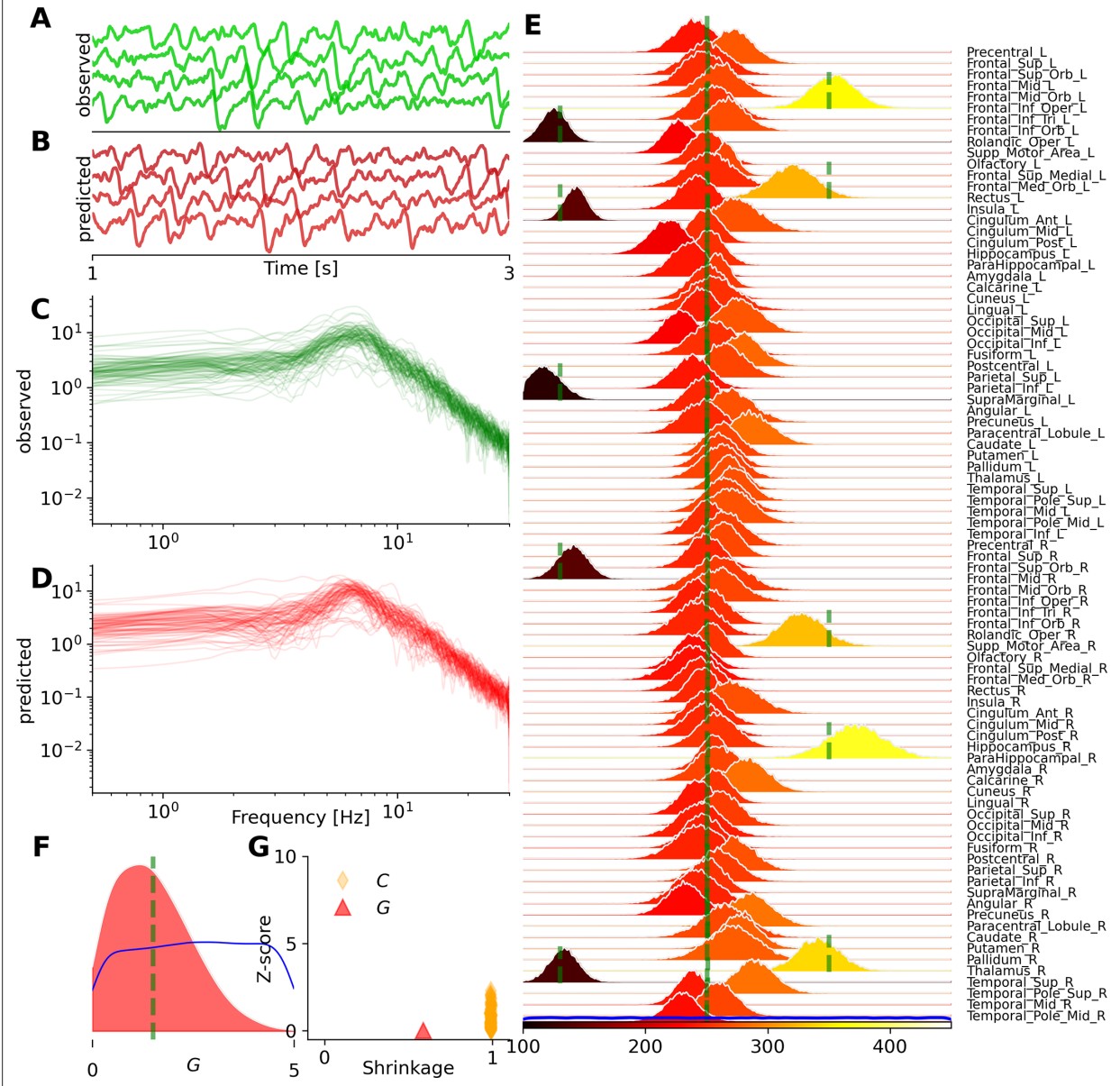

**Figure 4.** Bayesian inference on heterogeneous control parameters in the whole-brain network of the Jansen-Rit model. The set of inferred parameters is $\vec{\theta} = \{G, C_i\} \in \mathbb{R}^{89}$, with the global scaling parameter $G$ and average numbers of synapse between neural populations $C_i$ per region, given $i \in \{1, 2, ..., N_n = 88\}$ parcelled regions. Summary statistics of power spectrum density (PSD) were used for training, with a budget of 50 k simulations. (**A**) and (**B**) illustrate the observed and predicted neural activities, respectively. (**C**) and (**D**) show the observed and predicted data features, such as PSD. (**E**) and (**F**) display the posterior distribution of $C_i$ per region, and global coupling $G$, respectively. The ground truth and prior are represented by vertical green lines and a blue distribution, respectively. (**G**) shows the inference evaluation using the shrinkage and z-score of the estimated posterior distributions.

The online version of this article includes the following figure supplement(s) for figure 4:

**Figure supplement 1.** Inference on heterogeneous control parameters while excluding total power from the data features.

of high-dimensional heterogeneous parameters (given the ground truth, shown in green), leading to a strong correspondence between the observed and predicted PSD of EEG/MEG data. *Figure 4G* displays the shrinkage and z-score as the evaluation metrics, indicating an ideal Bayesian estimation for $C_i$ parameters, but not for the coupling parameter $G$. This occurred because the network input did not induce a significant change in the intrinsic frequency of activities at the regional level, resulting in diffuse uncertainty in its estimation for this model.

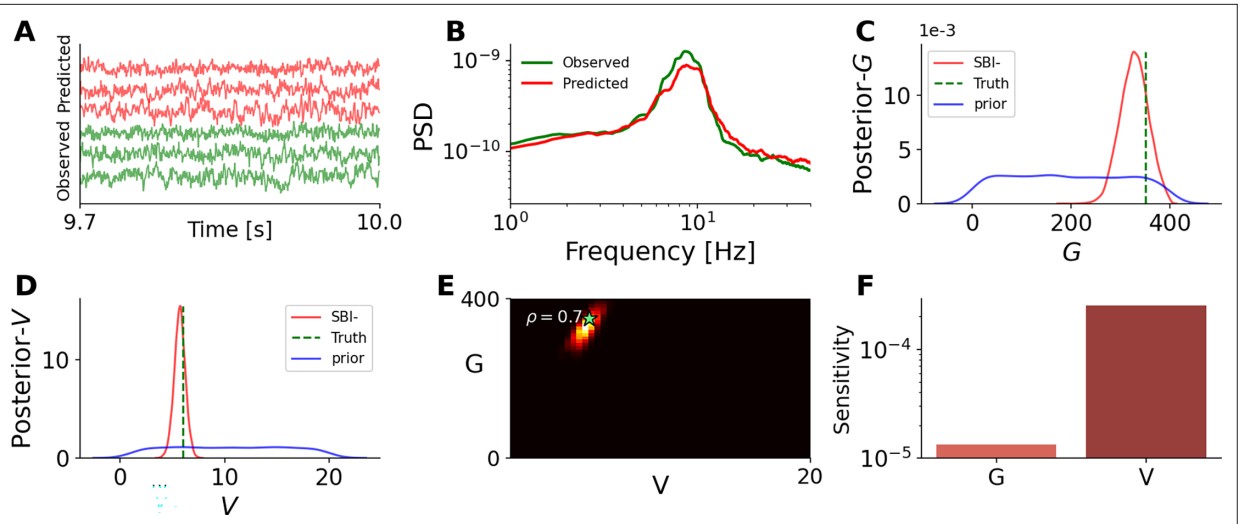

**Figure 5.** Bayesian inference on global scaling parameter $G$ and the averaged velocity $V$ of signal transmission using the whole-brain network model of Stuart-Landau oscillators. The set of estimated parameters is $\vec{\theta} = \{G, V\} \in \mathbb{R}^2$, and the summary statistics of PSD signals with a budget of 2 k simulations were used for training. (**A**) illustrates exemplary observed and predicted neural activities (in green and red, respectively). (**B**) shows the observed and predicted PSD signals (in green and red, respectively). (**C**) and (**D**) display the posterior distribution of averaged velocity $V$ and global coupling $G$, respectively. The true values and prior are shown as vertical green lines and a blue distribution, respectively. (**E**) shows the joint posterior distribution indicating a high correlation between posterior samples. (**F**) illustrates the sensitivity analysis based on the eigenvalues of the posterior distribution.

Note that relying on only alpha-peak while excluding other summary statistics, such as total power (i.e., area under the curve), leads to poor estimation of synaptic connections across brain regions (see *Figure 4—figure supplement 1*). This results in less accurate predictions of the PSD, with more dispersion in their amplitudes. This example demonstrates that VBI provides a valuable tool for hypothesis evaluation and improved insight into data features by uncertainty quantification and their impact on predictions.

## Whole-brain network of Stuart-Landau oscillators

To demonstrate efficient inference on the whole-brain time delay from EEG/MEG data, we used a whole-brain network model of coupled generic oscillators (see *Equation 3*). This model could establish a causal link between empirical spectral changes and the slower conduction velocities observed in multiple sclerosis patients, resulting from immune system attacks on the myelin sheath (*Wang et al., 2024*; *Mazzara et al., 2025*). The parameter set to estimate is $\vec{\theta} = \{G, V\} \in \mathbb{R}^2$, consisting of the global scaling parameter $G$ and the averaged velocity of signal transmission $V$. The training was performed using a budget of only 2 k simulations, which was sufficient due to the low dimensionality of the parameter space. *Figure 5A* illustrates the comparison between observed (in green) and predicted neural activities (in red). *Figure 5B* shows a close agreement between observed and predicted PSD signals, as the data feature used for training. *Figure 5C and D* provide visualizations of the posterior distributions for the averaged velocity $V$ and the global coupling $G$. In these panels, we can see a large shrinkage in the posterior (in red) from the uniform prior (in blue) centered around the true values (vertical green lines). Importantly, *Figure 5E* presenting the joint posterior distribution indicates a high correlation of $\rho = 0.7$ between parameters $G$ and $V$. This illustrates the advantage of Bayesian estimation in identifying statistical relationships between parameters, which helps to detect degeneracy among them. This is crucial for causal hypothesis evaluation and guiding conclusions in clinical settings. Finally, *Figure 5F* illustrates the sensitivity analysis (based on the eigenvalues of the posterior distribution), revealing that the posterior is more sensitive to changes in $V$ compared to $G$. This highlights the relative impact of these parameters on the model's posterior estimates.

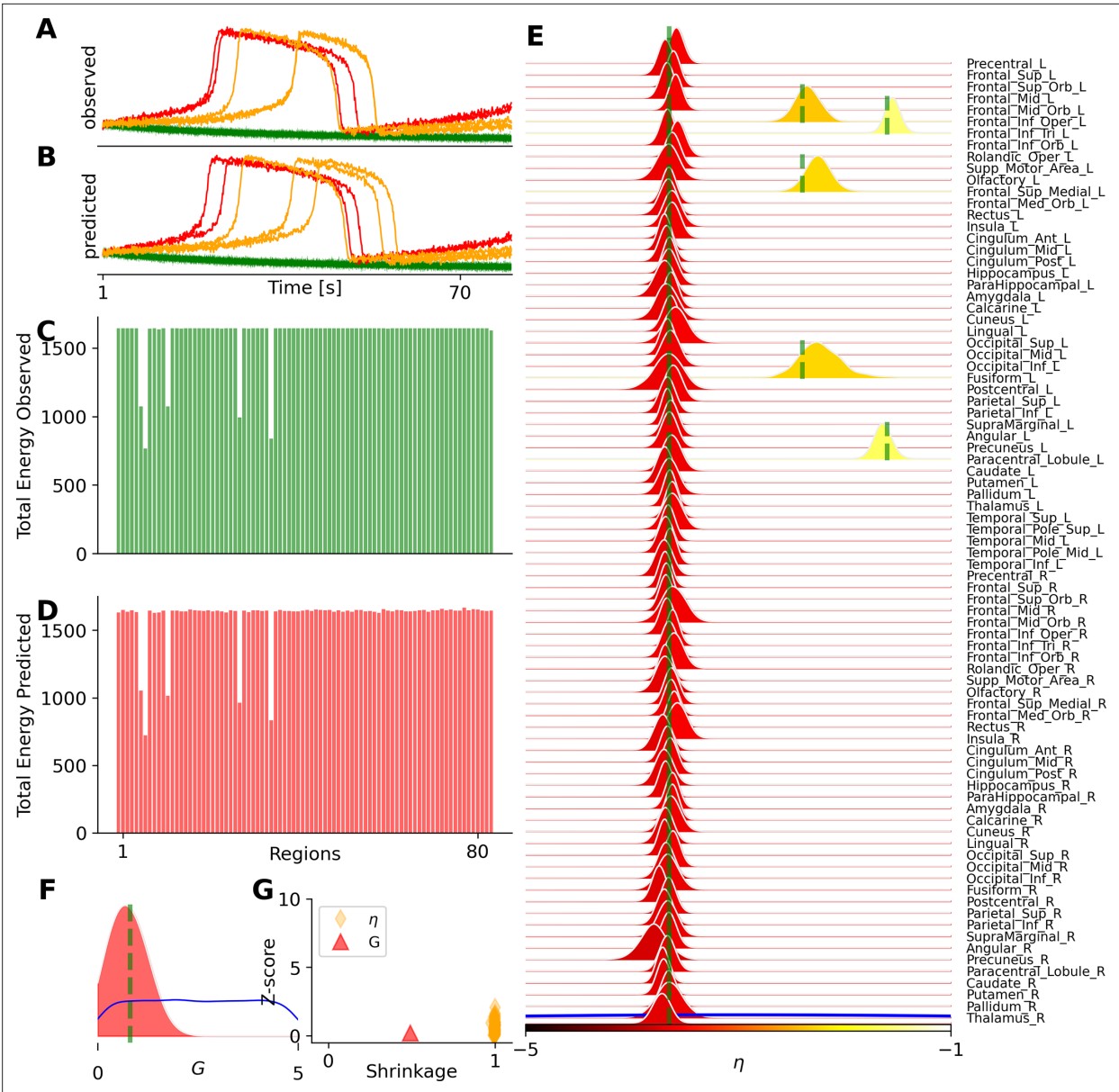

**Figure 6.** Bayesian inference on the spatial map of epileptogenicity across brain regions in the VEP model. The set of inferred parameters is $\vec{\theta} = \{G, \eta_i\} \in \mathbb{R}^{89}$, as the global scaling parameter and spatial map of epileptogenicity with $i \in \{1, 2, ..., N_n = 88\}$ parcelled regions. (**A**) The observed seizure envelope generated by the Epileptor model, given two regions as epileptogenic zones (in red) and three regions as propagation zones (in yellow), while the rest are healthy (in green). (**B**) The predicted seizure envelope, by training MAF model on a dataset containing 10 k simulations, using only the total power and seizure onset per region as the data features. (**C**) and (**D**) show the observed and predicted data features, respectively. (**E**) and (**F**) show the posterior distributions of heterogeneous control parameters $\eta_i$, and global coupling parameter $G$, respectively. (**G**) The posterior z-scores versus posterior shrinkages for estimated parameters.

The online version of this article includes the following figure supplement(s) for figure 6:

**Figure supplement 1.** Inference with slow time scale separation, $\tau = 10\,ms$.

**Figure supplement 2.** Evaluating the performance of estimation under different levels of additive noise.

**Figure supplement 3.** Evaluating the performance of estimation under different levels of dynamical noise.

## Whole-brain network of Epileptor model

Next, we demonstrate the inference on a whole-brain model of epilepsy spread, known as the Virtual Epileptic Patient (VEP; *Jirsa et al., 2017*; *Hashemi et al., 2020*), used to delineate the epileptogenic and propagation zone networks from the invasive sEEG recordings (see *Equation 4*). Here, we used a

large value for system time constant $\tau = 90\,ms$ to generate slow-fast dynamics in pathological areas, corresponding to seizure envelope at each brain region. *Figure 6* demonstrates the inference of the set of inferred parameters $\vec{\theta} = \{G, \eta_i\} \in \mathbb{R}^{89}$, with the global scaling parameter $G$ and spatial map of epileptogenicity $\eta_i$, given $i \in \{1, 2, ..., N_n = 88\}$ parcelled regions. *Figure 6A and B* show the observed and predicted envelope, respectively, at each brain region. Here, the whole brain regions are classified into two epileptogenic zones (in red, corresponding to high excitability), three propagation zones (in yellow, corresponding to excitability close to bifurcation), and the rest as healthy regions (in green, corresponding to low excitability). *Figure 6C and D* illustrate the observed and predicted data features as the total power energy per region, calculated as the area under the curve. Additionally, the seizure onset at each region was used as a data feature for training the MAF density estimator. From these panels, we observe accurate recovery of seizure envelopes in pathological regions. *Figure 6E and F* show that the posterior distribution of heterogeneous $\eta_i$, and global coupling parameter $G$, respectively, indicating 100% accurate recovery of the true values (in green). *Figure 6G* confirms the reliability and accuracy of the estimates through shrinkage and z-score diagnostics. With our efficient implementation, generating 10 k whole-brain simulations took less than a minute (using 10 CPU cores). The training took approximately 13 min to converge, while posterior sampling required only a few seconds. See *Figure 6—figure supplement 1* for a similar analysis with a slower time scale separation ($\tau = 10\,ms$). These results demonstrate an ideal and fast Bayesian estimation, despite the stiffness of equations in each region and the high dimensionality of the parameters. See *Figure 6—figure supplement 2* and *Figure 6—figure supplement 3* showing the accuracy and reliability of estimation under different levels of additive and dynamical noise. Note that for the VEP model, the total integration time is less than 100 ms, and due to the model's stable behavior and a large time step integration, the simulation cost is significantly lower compared to other whole-brain models.

## Whole-brain network of Montbrió model

Targeting the fMRI data, we demonstrate the inference on the whole-brain dynamics using Montbrió model (see *Equation 5*). *Figure 7* demonstrates the inference on heterogeneous control parameters of the Montbrió model, operating in a bistable regime. *Figure 7A and B* show the observed and predicted BOLD time series, respectively, while *Figure 7C and D* illustrate the observed and predicted data features, such as the static and dynamical functional connectivity matrices (FC and FCD, respectively). *Figure 7E and F* show the estimated posterior distributions of excitability $\eta_i$ per brain region, and the global coupling parameter $G$. *Figure 7G* displays the reliability and accuracy of estimation through the evaluation of posterior shrinkage and z-score (see *Equation 13* and *Equation 14*). See *Figure 7—figure supplement 1* for estimation over different configurations of the ground-truth values in this model.

Due to the large number of simulations for training and the informativeness of the data features (both spatio-temporal and functional data features), the results indicate that we achieved accurate parameter estimation and, consequently, a close agreement between the observed and predicted features of BOLD data. This required 500 k simulations for training, given the uniform priors $G \in \mathcal{U}(0, 1)$ and $\eta_i \in \mathcal{U}(-6, -3.5)$. After approximately 10 $h$ of training (of MAF density estimator), posterior sampling took only 1 min. Our results indicate that training the MAF model was two to four times faster than the NSF model. This showcase demonstrates the capability of VBI in inferring heterogeneous excitability, given the bistable brain dynamics, for fMRI studies. Note that removing the spatio-temporal features and considering only FC/FCD as the data features (see *Figure 7—figure supplement 2*) leads to poor estimation of the excitability parameter across brain regions (see *Figure 7—figure supplement 3*). Interestingly, accurate estimation of the only global coupling parameter, $\vec{\theta} = \{G\} \in \mathbb{R}^1$, from only FC/FCD requires around 100 simulations (see *Figure 7—figure supplement 4* and *Figure 7—figure supplement 5*). See *Figure 7—figure supplement 6* and *Figure 7—figure supplement 7* showing the accuracy and reliability of estimation under different levels of additive and dynamical noise.

## Whole-brain network of Wong-Wang model

Finally, in *Figure 8*, we show the inference on the so-called parameterized dynamics mean-field (pDMF) model, that is a whole-brain network model of the reduced Wong-Wang equation (see *Equation 6* and *Equation 7*), comprising 10 control parameters: the global scaling of connections $G$ and the linear coefficients $(a_w, b_w, c_w, a_I, b_I, c_I, a_\sigma, b_\sigma, c_\sigma) \in \mathbb{R}^9$. These parameters are introduced to reduce

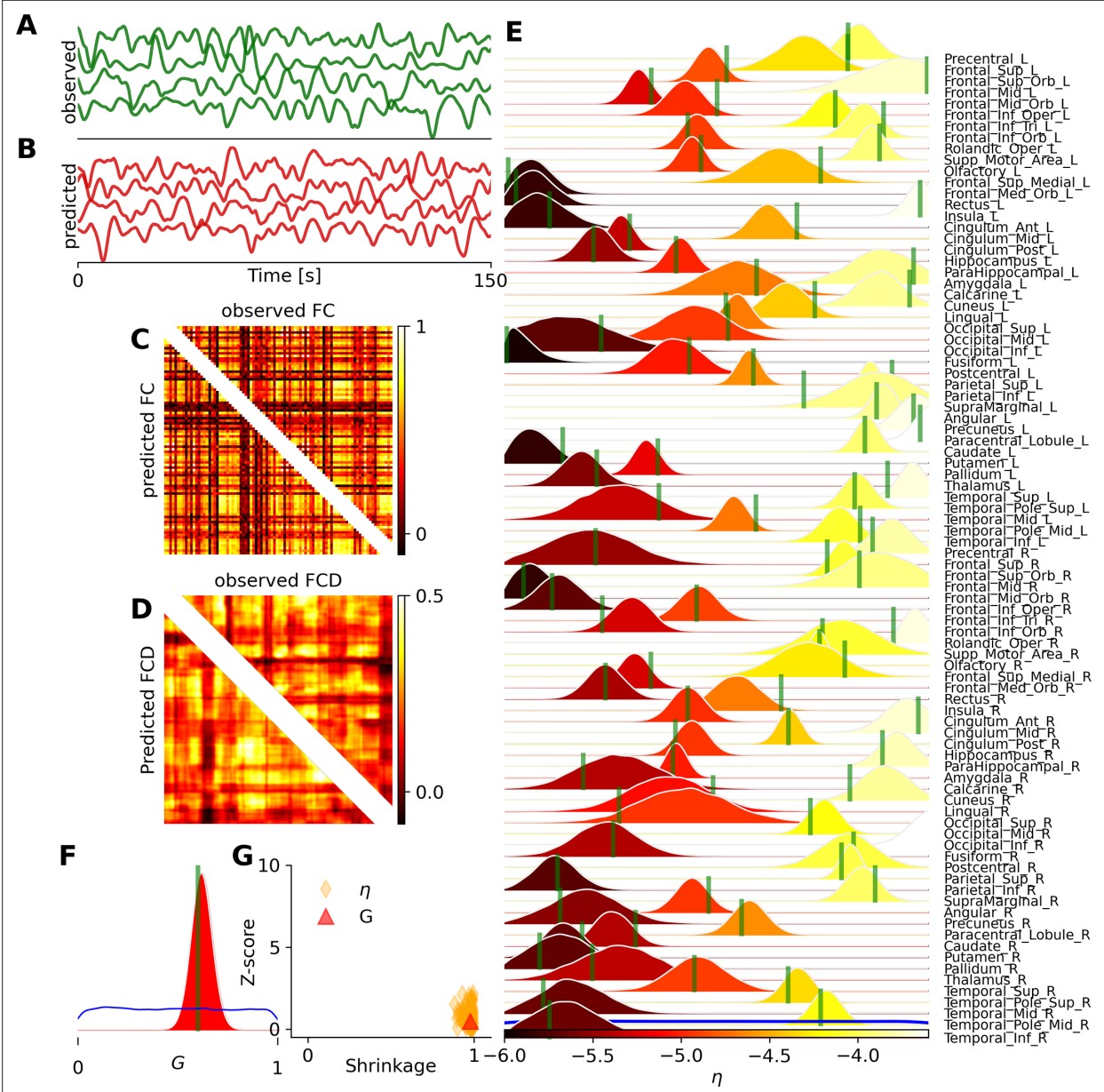

**Figure 7.** Bayesian inference on heterogeneous control parameters in the whole-brain dynamics using the Montbrió model. The set of inferred parameters is $\vec{\theta} = \{G, \eta_i\} \in \mathbb{R}^{89}$, as the global scaling parameter and excitability per region, with $i \in \{1, 2, ..., N_n = 88\}$ parcelled regions. VBI provides accurate and reliable posterior estimation using both spatio-temporal and functional data features for training, with a budget of 500 k simulations. (**A**) and (**B**) illustrate the observed and predicted BOLD signals, respectively. (**C**) and (**D**) show the observed (upper triangular) and predicted (lower triangular) data features (FC and FCD), respectively. (**E**) and (**F**) display the posterior distribution of excitability parameters $\eta_i$ per region, and global coupling $G$, respectively. The true values and prior are shown as vertical green lines and a blue distribution, respectively. (**G**) shows the inference evaluation by the shrinkage and z-score of the posterior distributions.

The online version of this article includes the following figure supplement(s) for figure 7:

**Figure supplement 1.** Inference across different configurations.

**Figure supplement 2.** Scatter plot of summary statistics used for training.

**Figure supplement 3.** Inference on heterogeneous control parameters using only FC/FCD matrices.

**Figure supplement 4.** Estimation of the coupling parameter using only functional data features via *VBI* simulator.

**Figure supplement 5.** Estimation of the coupling parameter using only functional data features via *TVB* simulator.

**Figure supplement 6.** Evaluating the performance of estimation under different levels of additive noise.

**Figure supplement 7.** Evaluating the performance of estimation under different levels of dynamical noise.

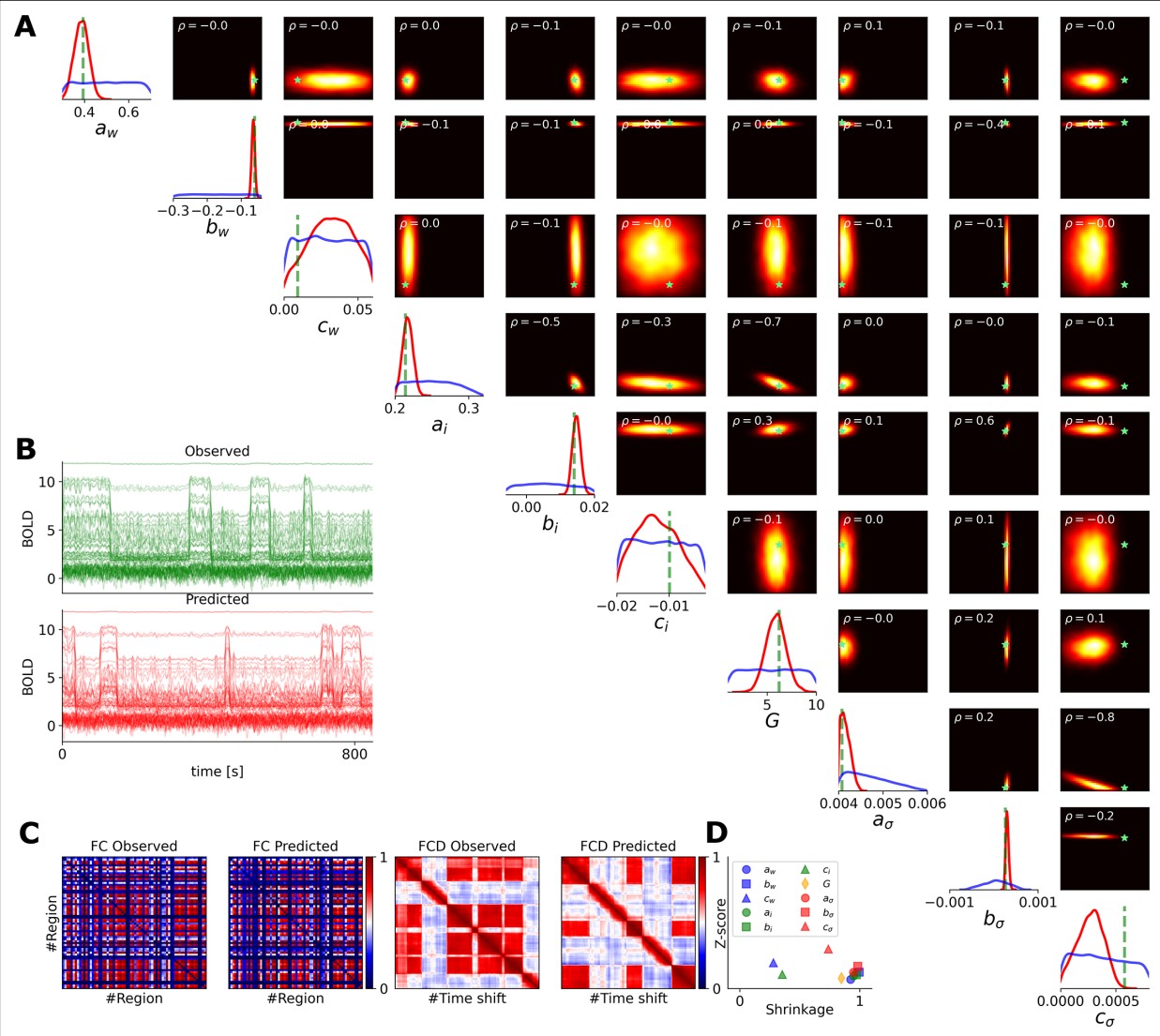

**Figure 8.** Bayesian inference on the parametric mean-field model of Wong-Wang (also known as pDMF model), with linear coefficients $(a_w, b_w, c_w, a_I, b_I, c_I, a_\sigma, b_\sigma, c_\sigma) \in \mathbb{R}^9$, reparameterizing the recurrent connection strength $w_i$, external input current $I_i$, and noise amplitude $\sigma_i$ for each region. Summary statistics of spatio-temporal and functional data features were used for training, with a budget of 200 k simulations. (**A**) The diagonal panels display the ground-true values (in green), the uniform prior (in blue), and the estimated posterior distributions (in red). The upper diagonal panels illustrate the joint posterior distributions between parameters, along with their correlation ($\rho$, in the upper left corners), and ground-truth values (green stars). High-probability areas are color-coded in yellow, while low-probability areas are shown in black. (**B**) The observed and predicted BOLD time series (in green and red, respectively). (**C**) The observed and predicted data features, such as FC/FCD matrices. (**D**) The inference evaluation by calculating the shrinkage and z-score of the estimated posterior distributions.

The online version of this article includes the following figure supplement(s) for figure 8:

**Figure supplement 1.** The distribution of observed and predicted data features in the pDMF model.

**Figure supplement 2.** The non-identifiability in estimating $G$ and $J$ in the pDMF model.

**Figure supplement 3.** Scatter plot of summary statistics used for training.

the dimension of whole-brain parameters as recurrent connection strength $w_i$, external input current $I_i$, and noise amplitude $\sigma_i$ for each region (in total, 264 parameters were reduced to 9 dimensions; see *Equation 7*). Here, we used summary statistics of both spatio-temporal and functional data features extracted from simulated BOLD data to train the MAF density estimator, with a budget of 200 k simulations. The training took around 160 min to converge, whereas posterior sampling took only a few seconds.

The diagonal panels in *Figure 8A* show estimated posterior distributions (in red), along with the prior (in blue), and true values (in green). The upper diagonal panels illustrate the joint posterior distributions between parameters (i.e. statistical dependency between parameters). *Figure 8B* illustrates the observed and predicted BOLD time series, generated by true and estimated parameters (in blue and red, respectively). From *Figure 8C*, we can see a close agreement between the observed and predicted data features (FC/FCD matrices). Note that due to the stochastic nature of the generative process, we do not expect an exact element-wise correspondence between these features, but rather a match in their summary statistics, such as the mean, variance, and higher order moments (see *Figure 8—figure supplement 1*). *Figure 8D* shows the posterior z-score versus shrinkage, indicating less accurate estimation for the coefficients $c_w$, $c_I$, and $c_\sigma$ compared to others, as they are not informed by anatomical data such as the T1w/T2w myelin map and the first FC principal gradient (see *Equation 7*). This showcase demonstrates the advantage of Bayesian inference over optimization in assessing the accuracy and reliability of parameter estimation, whether informed by anatomical data.

Note that in the whole-brain network of the Wong-Wang model, the global scaling parameter $G$ and synaptic coupling $J$ exhibit structural non-identifiability, meaning their combined effects on the system cannot be uniquely disentangled (see *Figure 8—figure supplement 2*, and *Figure 8—figure supplement 3*). This is evident in the parameter estimations corresponding to selected observations, where the posterior distributions appear diffuse. The joint posterior plots reveal a nonlinear dependency (banana shape) between $G$ and $J$, arising from their product in the neural mass equation (see *Equation 6*). Such a nonlinear relationship between parameters poses challenges for deriving causal conclusions, as often occurs in other neural mass models. This is a demonstration of how Bayesian inference facilitates causal hypothesis testing without requiring additional non-identifiability analysis.

## Discussion

This study introduces the VBI, a flexible and integrative toolkit designed to facilitate probabilistic inference on complex whole-brain dynamics using connectome-based models (forward problem) and SBI (inverse problem). The toolkit leverages high-performance programming languages (C++) and dynamic compilers (such as Python's JIT compiler), alongside the computational power of parallel processors (GPUs), to significantly enhance the speed and efficiency of simulations. Additionally, VBI integrates popular feature extraction libraries with parallel multiprocessing to efficiently convert simulated time series into low-dimensional summary statistics. Moreover, VBI incorporates state-of-the-art deep neural density estimators (such as MAF and NSF generative models) to estimate the posterior density of control parameters within whole-brain models given low-dimensional data features. Our results demonstrated the versatility and efficacy of the VBI toolkit across commonly used whole-brain network models, such as Wilson-Cowan, Jansen-Rit, Stuart-Landau, Epileptor, Montbrió, and Wong-Wang equations placed at each region. The ability to perform parallel and rapid simulations, coupled with a taxonomy of feature extraction, allows for detailed and accurate parameter estimation from associated neuroimaging modalities such as (s)EEG/MEG/fMRI. This is crucial for advancing our understanding of brain dynamics and the underlying mechanisms of various brain disorders. Overall, VBI represents a substantial improvement over alternative methods, offering a robust framework for both simulation and parameter estimation and contributing to the advancement of network neuroscience, potentially, to precision medicine.

The alternatives for parameter estimation include optimization techniques (*Hashemi et al., 2018*; *Wang et al., 2019*; *Kong et al., 2021*; *Cabral et al., 2022*; *Liu et al., 2023*), approximate Bayesian computation (ABC) method, and MCMC sampling (*Jha et al., 2022*). Optimization techniques are sensitive to the choice of the objective function (e.g. minimizing distance error or maximizing correlation) and do not provide estimates of uncertainty. Although multiple runs and thresholding can be used to address these issues, such methods often fall short in revealing relationships between parameters, such as identifying degeneracy, which is crucial for reliable causal inference. Alternatively, a technique known as ABC compares observed and simulated data using a distance measure based on summary statistics (*Beaumont et al., 2002*; *Beaumont, 2010*; *Sisson et al., 2018*). It is known that ABC methods suffer from the curse of dimensionality, and their performance also depends critically on the tolerance level in the accepted/rejected setting (*Gonçalves et al., 2020*; *Cranmer et al., 2020*). The self-tuning variants of MCMC sampling have also been used for model inversion at the whole-brain level (*Hashemi et al., 2020*; *Hashemi et al., 2021*). Although MCMC is unbiased and exact with

infinite runs, it can be computationally prohibitive, and sophisticated reparameterization methods are often required to facilitate convergence at whole-brain level (*Jha et al., 2022*; *Baldy et al., 2023*). This becomes more challenging for gradient-based MCMC algorithms, due to the bistability and stiffness of neural mass models. Tailored to Bayes' rule, SBI sidesteps these issues by relying on expressive deep neural density estimators (such as MAF and NSF) on low-dimensional data features to efficiently approximate the posterior distribution of model parameters. Taking spiking neurons as generative models, this approach has demonstrated superior performance compared to alternative methods, as it does not require model or data features to be differentiable (*Gonçalves et al., 2020*; *Baldy et al., 2024*).

In previous studies, we demonstrated the effectiveness of SBI on virtual brain models of neurological (*Hashemi et al., 2023*; *Wang et al., 2024*; *Mazzara et al., 2025*; *Mazzara et al., 2025*), and neurodegenerative diseases (*Hashemi et al., 2024*; *Yalçınkaya et al., 2023*; *Hashemi et al., 2025*) as well as focal intervention (*Rabuffo et al., 2025*) and healthy aging (*Lavanga et al., 2023*). In this work, we extended this probabilistic methodology to encompass a broader range of whole-brain network models, highlighting its flexibility and scalability in leveraging diverse computational resources, from CPUs/GPUs to high-performance computing facilities.

Our results indicated that the VBI toolkit effectively estimates posterior distributions of control parameters in whole-brain network modeling, offering a deeper understanding of the mechanisms underlying brain activity. For example, using the Montbrió and Wong-Wang models, we achieved a close match between observed and predicted FC/FCD matrices derived from BOLD time series (*Figures 7 and 8*). Additionally, the Jansen-Rit and Stuart-Landau models provided accurate inferences of PSD from neural activity (*Figures 4 and 5*), while the Epileptor model precisely captured the spread of seizure envelopes (*Figure 6*). These results underscore the toolkit's capability to manage complex, high-dimensional data with precision. Uncertainty quantification using VBI can illuminate and combine the informativeness of data features (e.g., FC/FCD) and reveal the causal drivers behind interventions (*Lavanga et al., 2023*; *Hashemi et al., 2024*; *Rabuffo et al., 2025*). This adaptability ensures that VBI can be applied across various (source-localized) neuroimaging modalities, accommodating different computational capabilities and research needs.

Note that there is no specific rule for determining the optimal number of simulations required for training. In general, a larger number of simulations, depending on the available computational resources, tends to improve the quality of posterior estimation. However, when using synthetic data, we can monitor the z-score and posterior shrinkage to assess the accuracy and reliability of the inferred parameters. This also critically depends on the parameter dimensionality. For instance, in estimating only global coupling parameter, a maximum of 300 simulations was used, demonstrating accurate estimation across models and different configurations (see *Figure 2—figure supplement 3*), except for the Jansen-Rit model, where coupling did not induce a significant change in the intrinsic frequency of regional activity. Importantly, the choice of data features is critical, and some factors (e.g. that lead to inaccurate feature calculation) can lead to the collapse of this method. For instance, high noise levels in observations or dynamical noise can compromise the accurate calculation of data features, undermining the inference process (see *Figure 6—figure supplement 2*, *Figure 6—figure supplement 3*, *Figure 7—figure supplement 6*, *Figure 7—figure supplement 7*). Identifying the set of low-dimensional data features that are relevant to the control parameters for each case study is another challenge in effectively applying SBI. Nevertheless, the uncertainty of the posterior informs us about the predictive power of these features. Statistical moments of time series could be effective candidates for most models. However, this poses a formidable challenge for inference from empirical data, as certain moments, such as the mean and variance, may be lost during preprocessing steps. The hyperparameter and noise estimation can also be challenging for SBI. Moreover, there is no established rule for determining the number of simulations for training, aside from relying on z-score values during in silico testing, as it depends on available computational resources.

Various sequential methods, such as SNPE (*Greenberg et al., 2019*), SNLE (*Lueckmann et al., 2019*), and SNRE (*Durkan et al., 2020*), have been proposed to reduce computational costs of SBI by iteratively refining the fit to specific targets. These approaches aim for more precise parameter estimation by progressively adjusting the model based on each new data set or subset, potentially enhancing the accuracy of the fit at the reduced computational effort. The choice of method depends on the specific characteristics and requirements of the problem being addressed (*Lueckmann et al.,*

*2021*). Our previous study indicates that for inferring whole-brain dynamics of epilepsy spread, the SNPE method outperforms alternative approaches (*Hashemi et al., 2023*). Nevertheless, sequential methods can become unstable, with simulators potentially diverging and causing probability mass to leak into regions that lack prior support (*Hashemi et al., 2023*). In this study, we used single-round training to benefit from an amortization strategy. This approach brings the costs of simulation and network training upfront, enabling inference on new data to be performed rapidly (within seconds). This strategy facilitates personalized inference at the subject level, as the generative model is tailored by the SC matrix, thereby allowing for rapid hypothesis evaluation specific to each subject (e.g. in delineating the epileptogenic and propagation zones). Note that model comparison across different configurations or model structures, as well established in dynamic causal modeling (*Penny et al., 2004*; *Penny, 2012*; *Baldy et al., 2025*), has yet to be explored in this context.

Deep learning algorithms are increasingly gaining traction in the context of whole-brain modeling. The VBI toolkit leverages a class of deep generative models, called Normalizing Flows (NFs; *Rezende and Mohamed, 2015*; *Kobyzev et al., 2019*), to model probability distributions given samples drawn from those distributions. Using NFs, a base probability distribution (e.g. a standard normal) is transformed into any complex distribution (potentially multi-modal) through a sequence of invertible transformations. Variational autoencoders (VAEs; *Kingma et al., 2014*; *Doersch, 2016*) are a class of deep generative models to encode data into a latent space and then decode it back to reconstruct the original data. Recently, *Sip et al., 2023* introduced a method using VAEs for nonlinear dynamical system identification, enabling the inference of neural mass models and region- and subject-specific parameters from functional data. VAEs have also been employed for dimensionality reduction of whole-brain functional connectivity (*Perl et al., 2023b*), and to investigate various pathologies and their severity by analyzing the evolution of trajectories within a low-dimensional latent space (*Perl et al., 2023a*). Additionally, Generative Adversarial Networks (GANs; *Goodfellow et al., 2014*; *Creswell et al., 2018*) have demonstrated remarkable success in mapping latent space to data space by learning a manifold induced from a base density (*Liu et al., 2021*). This method merits further exploration within the context of whole-brain dynamics. To fully harness the potential of deep generative models in large-scale brain network modeling, integrating VAEs and GANs into the VBI framework would be beneficial. This will elucidate their strengths and limitations within this context and guide future advancements in the field.

In summary, VBI offers fast simulations, taxonomy of feature extraction, and deep generative models, making it a versatile tool for model-based inference from different neuroimaging modalities, helping researchers to explore brain (dys)functioning in greater depth. This advancement not only enhances our theoretical understanding but also holds promise for practical applications in diagnosing and treating neurological conditions.

## Materials and methods
### The virtual brain models

To build a virtual brain model (see *Figure 1*), the process begins with parcellating the brain into regions using anatomical data, typically derived from T1-MRI scans. Each region, represented as nodes in the network, is then equipped with a neural mass model to simulate the collective behavior of neurons within that area. These nodes are interconnected using a structural connectivity (SC) matrix, typically obtained from diffusion-weighted magnetic resonance imaging (DW-MRI). The connectome was built with TVB-specific reconstruction pipeline using generally available neuroimaging software (*Schirner et al., 2015*). The entire network of interconnected nodes is then simulated using neuroinformatic tools, such as The Virtual Brain (TVB; *Sanz-Leon et al., 2015*), generating neural activities at the source level. However, neural sources are not directly observable in real-world experiments, and a projection needs to be established to transform the simulated neural activity into empirically measurable quantities, such as (s)EEG, MEG, and fMRI. This approach offers insights into both normal brain function and neurological disorders (*Hashemi et al., 2025*). In the following, we describe commonly used whole-brain network models at the source level, which can be mapped to different types of neuroimaging recordings. Note that each model represents one of many possible variants used in the literature, and the choice of model often depends on the specific research question, the spatial and temporal resolution of the available data, and the desired level of biological or mathematical detail.

## Wilson-Cowan model

The Wilson-Cowan model (*Wilson and Cowan, 1972*) is a seminal neural mass model that describes the dynamics of connected excitatory and inhibitory neural populations, at cortical microcolumn level. It has been widely used to understand the collective behavior of neurons and simulate neural activities recorded by methods such as local field potentials (LFPs) and EEG. The model effectively captures phenomena such as oscillations, wave propagation, pattern formation in neural tissue, and responses to external stimuli, offering insights into various brain (dys)functions, particularly in Parkinson's disease (*Duchet et al., 2021*; *Sermon et al., 2023*).

The Wilson-Cowan model describes the temporal evolution of the mean firing rates of excitatory ($E$) and inhibitory ($I$) populations using nonlinear differential equations. Each population's activity is governed by a balance of self-excitation, cross-inhibition, external inputs, and network interactions through long-range coupling. The nonlinearity arises from a sigmoidal transfer function $S_{i,e}(x)$, which maps the total synaptic input to the firing rate, capturing saturation effects and thresholds in neural response. In the whole-brain network extension, each neural population at node $i$ receives input from other nodes via a weighted connectivity matrix, allowing the study of large-scale brain dynamics and spatial pattern formation (*Wilson and Cowan, 1972*; *Wilson and Cowan, 1973*; *Daffertshofer and van Wijk, 2011*):

**Table 1.** Parameter descriptions for capturing whole-brain dynamics using the Wilson-Cowan neural mass model.

| Parameter | Description | Value | Prior |
|---|---|---|---|
| $c_{ee}$ | Excitatory to excitatory synaptic strength | 16.0 | |
| $c_{ei}$ | Inhibitory to excitatory synaptic strength | 12.0 | |
| $c_{ie}$ | Excitatory to inhibitory synaptic strength | 15.0 | |
| $c_{ii}$ | Inhibitory to inhibitory synaptic strength | 3.0 | |
| $\tau_e$ | Time constant of excitatory population | 8.0 | |
| $\tau_i$ | Time constant of inhibitory population | 8.0 | |
| $a_e$ | Sigmoid slope for excitatory population | 1.3 | |
| $a_i$ | Sigmoid slope for inhibitory population | 2.0 | |
| $b_e$ | Sigmoid threshold for excitatory population | 4.0 | |
| $b_i$ | Sigmoid threshold for inhibitory population | 3.7 | |
| $c_e$ | Maximum output of sigmoid for excitatory population | 1.0 | |
| $c_i$ | Maximum output of sigmoid for inhibitory population | 1.0 | |
| $\theta_e$ | Firing threshold for excitatory population | 0.0 | |
| $\theta_i$ | Firing threshold for inhibitory population | 0.0 | |
| $r_e$ | Refractoriness of excitatory population | 1.0 | |
| $r_i$ | Refractoriness of inhibitory population | 1.0 | |
| $k_e$ | Scaling constant for excitatory output | 0.994 | |
| $k_i$ | Scaling constant for inhibitory output | 0.999 | |
| $\alpha_e$ | Gain of excitatory population | 1.0 | |
| $\alpha_i$ | Gain of inhibitory population | 1.0 | |
| $P$ | External input to excitatory population | 0.0 | $\mathcal{U}(0,3)$ |
| $Q$ | External input to inhibitory population | 0.0 | $\mathcal{U}(0,3)$ |
| $g_e$ | Global coupling strength (excitatory) | 0.0 | $\mathcal{U}(0,1)$ |
| $g_i$ | Global coupling strength (inhibitory) | 0.0 | $\mathcal{U}(0,1)$ |
| $\sigma$ | Standard deviation of Gaussian noise | 0.005 | |

$$\tau_e \frac{dE_k}{dt} = -E_k + (k_e - r_e E_k) \cdot S_e \left( \alpha_e \left( c_{ee} E_k - c_{ei} I_k + P_k - \theta_e + g_e \sum_l \text{SC}_{kl} E_l \right) \right) + \sigma \xi_k(t),$$

$$\tau_i \frac{dI_k}{dt} = -I_k + (k_i - r_i I_k) \cdot S_i \left( \alpha_i \left( c_{ie} E_k - c_{ii} I_k + Q_k - \theta_i + g_i \sum_l \text{SC}_{kl} I_l \right) \right) + \sigma \xi_k(t),$$

$$S_k(x) = \begin{cases} c_k \left( \dfrac{1}{1 + e^{-a_k(x - b_k)}} - \dfrac{1}{1 + e^{a_k b_k}} \right), & \text{if shifted,} \\ \dfrac{c_k}{1 + e^{-a_k(x - b_k)}}, & \text{otherwise,} \quad k = i, e \end{cases}$$

(1)

which incorporates both local dynamics and global interactions, modulated by coupling strengths and synaptic weights. Here, $\text{SC}_{kl}$ is an element of the (non)symmetric structural connectivity matrix and is nonzero if there is a connection between regions $k$ and $l$. The nominal parameter values and the prior range for the target parameters are summarized in *Table 1*.

## Jansen-Rit model

The Jansen-Rit neural mass model has been widely used to simulate physiological signals from various recording methods like intracranial LFPs and scalp MEG/EEG recordings. For example, it has been shown to recreate responses similar to evoked-related potentials after a series of impulse stimulations (*David and Friston, 2003*; *David et al., 2006*), generating high-alpha and low-beta oscillations, with added recurrent inhibitory connections and spike-rate modulation (*Moran et al., 2007*), and also seizure patterns similar to those seen in temporal lobe epilepsy (*Wendling et al., 2001*). This biologically motivated model comprises three main populations of neurons: excitatory pyramidal neurons, inhibitory interneurons, and excitatory interneurons. These populations interact with each other through synaptic connections, forming a feedback loop that produces oscillatory activity governed by a set of nonlinear ordinary differential equations (*Jansen and Rit, 1995*; *David and Friston, 2003*; *Daffertshofer and van Wijk, 2022*):

**Table 2.** Parameter descriptions for capturing whole-brain dynamics using Jansen-Rit neural mass model.
EP: excitatory populations, IP: inhibitory populations, PSP: post synaptic potential, PSPA: post synaptic potential amplitude.

| Parameters | Description | Value | Prior |
|---|---|---|---|
| $A$ | Excitatory PSPA | 3.25 mV | |
| $B$ | Inhibitory PSPA | 22 mV | |
| $1/a$ | Time constant of excitatory PSP | $a = 100\,s^{-1}$ | |
| $1/b$ | Time constant of inhibitory PSP | $b = 50\,s^{-1}$ | |
| $C_1, C_2$ | Average numbers of synapses between EP | $1C, 0.8C$ | |
| $C_3, C_4$ | Average numbers of synapses between IP | $0.25C$ | |
| $v_{max}$ | Maximum firing rate | 5 Hz | |
| $v_0$ | Potential at half of maximum firing rate | 6 mV | |
| $r$ | Slope of sigmoid function at $v_0$ | $0.56\,mV^{-1}$ | |
| $C$ | Average numbers of synapses between neural populations | 135 | $\mathcal{U}(100, 500)$ |
| $G$ | Scaling the strength of network connections | 1.5 | $\mathcal{U}(0, 5)$ |

$$\dot{y}_{0i}(t) = y_{3i}(t); \qquad \dot{y}_{1i}(t) = y_{4i}(t); \qquad \dot{y}_{2i}(t) = y_{5i}(t)$$

$$\dot{y}_{3i}(t) = A\,a\,S\big(y_{1i}(t) - y_{2i}(t)\big) - 2a\,y_{3i}(t) - a^2 y_{0i}(t)$$

$$\dot{y}_{4i}(t) = A\,a\big(P(t) + C_2 S(C_1 y_{0i}(t)) + G\,H_i\big) - 2a\,y_{4i}(t) - a^2 y_{1i}(t)$$

$$\dot{y}_{5i}(t) = B\,b\big(C_4 S(C_3 y_{0i}(t))\big) - 2b\,y_{5i}(t) - b^2 y_{2i}(t) \qquad (2)$$

$$S(\nu) = \frac{v_{\max}}{1 + \exp\big(r(v_0 - \nu)\big)}$$

$$H_i = \sum_{j=1}^{N} SC_{ij}\,S\big(y_{1j} - y_{2j}\big)$$

where $y_{0i}$, $y_{1i}$, and $y_{2i}$ denote the average membrane potentials of pyramidal cells, excitatory interneurons, and inhibitory interneurons, respectively. Their corresponding time derivatives, $y_{3i}(t)$, $y_{4i}(t)$, and $y_{5i}(t)$, represent the rates of change of these membrane potentials. $P(t)$ also represents an external input current. The sigmoid function $S(x)$ maps the average membrane potential of neurons to their mean action potential firing rate. SC is a normalized structural connectivity matrix. The model's output at $i_{th}$ region corresponds to the membrane potential of pyramidal cells and is given by $y_{1i} - y_{2i}$. The nominal parameter values and the prior range for the target parameters are summarized in **Table 2**.

## Stuart-Landau oscillator

The Stuart-Landau oscillator (**Selivanov et al., 2012**) is a generic mathematical model used to describe oscillatory phenomena, particularly those near a Hopf bifurcation, which is often employed to study the nonlinear dynamics of neural activity (**Deco et al., 2017**; **Petkoski and Jirsa, 2019**; **Cabral et al., 2022**; **Wang et al., 2024**). One approach uses this model to capture slow hemodynamic changes in BOLD signal (**Deco et al., 2017**), while others apply it to model fast neuronal dynamics, which can be linked directly to EEG/MEG data (**Petkoski and Jirsa, 2019**; **Cabral et al., 2022**; **Wang et al., 2024**). Note that this is a phenomenological framework, and both applications operate on completely different time scales.

In the network, each brain region, characterized by an autonomous Stuart-Landau oscillator, can exhibit either damped or limit-cycle oscillations depending on the bifurcation parameter $a$. If $a < 0$, the system shows damped oscillations, similar to a pendulum under friction. In this regime, the system, when subjected to perturbation, relaxes back to its stable fixed point through damped oscillations with an angular frequency $\omega_0$. The rate of amplitude damping is determined by $|a|$. Conversely, if $a > 0$, the system supports limit cycle solutions, allowing for self-sustained oscillations even in the absence of external noise. At a critical value of $a = 0$, the system undergoes a Hopf bifurcation, that is small changes in parameters can lead to large variations in the system's behavior.

Using whole-brain network modeling of EEG/MEG data (**Sorrentino et al., 2024**; **Cabral et al., 2022**), the oscillators are interconnected via white-matter pathways, with coupling strengths specified by subject-specific DTI fiber counts, that is elements of SC matrix. This adjacency matrix is then scaled by a global coupling parameter $G$. Note that coupling between regions accounts for finite conduction times, which are often estimated by dividing the Euclidean distances between nodes by an average conduction velocity $T_{jk} = d_{jk}/v$. Knowing the personalized time-delays (**Lemaréchal et al., 2022**; **Sorrentino et al., 2022**), we can use the distance as a proxy, assuming a constant propagation velocity. The distance itself can be defined as either the length of the tracts or the Euclidean distance. Taking this into account, the activity of each region is given by a set of complex differential equations:

$$\dot{Z}_j = Z_j(a + i\omega_j - |Z_j|^2) + G\sum_{k=1}^{N} SC_{jk}[Z_k(t - T_{jk}) - Z_j(t)] + \sigma(\xi_{j1}(t) + i\xi_{j2}(t)) \qquad (3)$$

where $Z$ is a complex variable, and $Re[Z(t)]$ is the corresponding time series. In this particular realization, each region has a natural frequency of 40 Hz ($\omega_j = \omega_0 = 2\pi \cdot 40\ rad/s$), motivated by empirical studies demonstrating the emergence of gamma oscillations from the balance of excitation and inhibition, playing a role in local circuit computations (**Funk and Epstein, 2004**).

**Table 3.** Parameter descriptions for capturing whole-brain dynamics using Stuart-Landau oscillator.

| Parameters | Description | Value | Prior |
|---|---|---|---|
| $a$ | Bifurcation parameter | -5 | |
| $\omega_i$ | Natural angular frequency | $2\pi \cdot 40 \, \text{rad/s}$ | |
| $\sigma$ | Noise factor | $10^{-4}$ | |
| $v$ | Average conduction velocity | 6.0 m/s | $\mathcal{U}(1, 30)$ |
| $G$ | Global coupling parameter | 350 | $\mathcal{U}(0, 400)$ |

In this study, for the sake of simplicity, a common cortico-cortical conduction velocity is estimated, that is the distance-dependent average velocities $v_{jk} = v$. We also consider $a = -5$, capturing the highly variable amplitude envelope of gamma oscillations as reported in experimental recordings (*Buzsáki and Wang, 2012*; *Cabral et al., 2022*). This choice also best reflects the slowest decay time constants of GABA$_B$ inhibitory receptors–approximately 1 s (*Schnitzler and Gross, 2005*). A Gaussian noise (here denoted by $\xi_i$) with an intensity of $\sigma = 10^{-4}$ is added to each oscillator to mimic stochastic fluctuations. The nominal parameter values and the prior range for the target parameters are summarized in *Table 3*.

## Epileptor model

In personalized whole-brain network modeling of epilepsy spread (*Jirsa et al., 2017*), the dynamics of each brain region are governed by the Epileptor model (*Jirsa et al., 2014*). The Epileptor model provides a comprehensive description of epileptic seizures, encompassing the complete taxonomy of system bifurcations to simultaneously reproduce the dynamics of seizure onset, progression, and termination (*Saggio et al., 2020*). The full Epileptor model comprises five state variables that couple two oscillatory dynamical systems operating on three different time scales (*Jirsa et al., 2014*). Then, motivated by Synergetic theory (*Haken, 1977*; *Jirsa and Haken, 1997*) and under time-scale separation (*Proix et al., 2014*), the fast variables rapidly collapse on the slow manifold *McIntosh and Jirsa, 2019*, whose dynamics is governed by the slow variable. This adiabatic approximation yields the 2D reduction of whole-brain model of epilepsy spread, also known as the VEP, as follows:

$$\dot{x}_i = 1 - x_i^3 - 2x_i^2 - z_i + I_i$$

$$\dot{z}_i = \frac{1}{\tau}(4(x_i - \eta_i) - z_i - G \sum_{j=1}^{N} \text{SC}_{ij}(x_j - x_i)),$$

(4)

where and indicate the fast and slow variables corresponding to brain region, respectively, and the set of unknowns is the spatial map of epileptogenicity to be estimated. SC is a normalized structural connectivity matrix. In real-world epilepsy applications (*Hashemi et al., 2023*; *Hashemi et al., 2021*; *Wang et al., 2023b*), we compute the envelope function from sEEG data to perform inference. The nominal parameter values and the prior range for the target parameters are summarized in *Table 4*.

**Table 4.** Parameter descriptions for capturing whole-brain dynamics using 2D Epileptor neural mass model.

| Parameter | Description | Value | Prior |
|---|---|---|---|
| $I$ | Input electric current | 3.1 | |
| $\tau$ | System time constant | 90ms | |
| $\eta_i$ | Spatial map of epileptogenicity | –3.65 | $\mathcal{U}(-5, -1)$ |
| $G$ | Global scaling factor on network connections | 1.0 | $\mathcal{U}(0, 2)$ |

## Montbrió model

The exact macroscopic dynamics of a specific brain region (represented as a node in the network) can be analytically derived in thermodynamic limit of infinitely all-to-all coupled spiking neurons (**Montbrió et al., 2015**) or Θ neuron representation (**Byrne et al., 2020**). By assuming a Lorentzian distribution on excitabilities in large ensembles of quadratic integrate-and-fire neurons with synaptic weights $J$ and a half-width $\Delta$ centered at $\eta$, the macroscopic dynamics has been derived in terms of the collective firing activity and mean membrane potential (**Montbrió et al., 2015**). Then, by coupling the brain regions via an additive current (e.g. in the average membrane potential equations), the dynamics of the whole-brain network can be described as follows (**Rabuffo et al., 2025**):

$$
\begin{aligned}
\tau \dot{r}_i(t) &= 2\, r_i(t)\, v_i(t) + \frac{\Delta}{\pi \tau} \\
\tau \dot{v}_i(t) &= v_i^2(t) - \left(\pi \tau r_i(t)\right)^2 + J\tau r_i(t) + \eta + G \sum_{j=1}^{N} \mathrm{SC}_{ij} r_j(t) + I_{\mathrm{stim}}(t) + \xi(t)
\end{aligned}
\tag{5}
$$

where $v_i$ and $r_i$ are the average membrane potential and firing rate, respectively, at the $i_{th}$ brain region, and parameter $G$ is the network scaling parameter that modulates the overall impact of brain connectivity on the state dynamics. The $\mathrm{SC}_{ij}$ denotes the connection weight between $i_{th}$ and $j-th$ regions, and the dynamical noise $\xi(t) \sim \mathcal{N}(0, \sigma^2)$ follows a Gaussian distribution with mean zero and variance $\sigma^2$.

The model parameters are tuned so that each decoupled node is in a bistable regime, exhibiting a down-state stable fixed point (low-firing rate) and an up-state stable focus (high-firing rate) in the phase space (**Montbrió et al., 2015**; **Baldy et al., 2024**). Bistability is a fundamental property of regional brain dynamics to ensure a switching behavior in the data (e.g. to generate FCD), that has been recognized as representative of realistic dynamics observed empirically (**Rabuffo et al., 2021**; **Breyton et al., 2023**; **Fousek et al., 2024**; **Rabuffo et al., 2025**). The solution of the coupled system yields a neuroelectric dataset that describes the evolution of the variables $(r_i(t), v_i(t))$ in each brain region $i$, providing measures of macroscopic activity. The surrogate BOLD activity for each region is then derived by filtering this activity through the Balloon-Windkessel model (**Friston et al., 2000**). The input current $I_{stim}$ represents the stimulation to selected brain regions, which increase the basin of attraction of the up-state in comparison to the down-state, while the fixed points move farther apart (**Rabuffo et al., 2021**; **Breyton et al., 2023**; **Hashemi et al., 2024**; **Rabuffo et al., 2025**). The nominal parameter values and the prior range for the target parameters are summarized in **Table 5**.

## Wong-Wang model

Another commonly used whole-brain model for simulation of neural activity is the so-called parameterized dynamics mean-field (pDMF) model (**Hansen et al., 2015**; **Kong et al., 2021**; **Deco et al., 2013**). At each region, it comprises a simplified system of two nonlinear coupled differential equations, motivated by the attractor network model, which integrates sensory information over time to make perceptual decisions, known as Wong-Wang model (**Wong and Wang, 2006**). This biophysically realistic cortical network model of decision making then has been simplified further into a single-population model (**Deco et al., 2013**), which has been widely used to understand the mechanisms underpinning brain resting state dynamics (**Kong et al., 2021**; **Deco et al., 2021**; **Zhang et al.,**

**Table 5.** Parameter descriptions for capturing whole-brain dynamics using Montbrió model.

| Parameter | Description | Nominal value | Prior |
|---|---|---|---|
| $\tau$ | Characteristic time constant | 1 ms | |
| $J$ | Synaptic weight | 14.5ms$^{-1}$ | |
| $\Delta$ | Spread of the heterogeneous noise distribution | 0.7ms$^{-1}$ | |
| $I_{\mathrm{stim}}(t)$ | Input current representing stimulation | 0.0 | |
| $\sigma$ | Gaussian noise variance | 0.037 | |
| $\eta$ | Excitability | –4.6 | $\mathcal{U}(-6, -3.5)$ |
| $G$ | Scaling the strength of network connections | 0.56 | $\mathcal{U}(0, 1)$ |

**Table 6.** Parameter descriptions for capturing whole-brain dynamics using the Wong-Wang model.

| Parameter | Description | Value | Prior |
|---|---|---|---|
| $a$ | Max feeding rate of $H(x)$ | 270 n/C | |
| $b$ | Half saturation of $H(x)$ | 108 Hz | |
| $d$ | Control the steepness of curve of $H(x)$ | 0.154 s | |
| $\gamma$ | Kinetic parameter | 0.641/1000 | |
| $\tau_s$ | Synaptic time constant | 100 ms | |
| $J$ | Synaptic coupling | 0.2609 nA | |
| $w$ | Local excitatory recurrence | 0.6 | $\mathcal{U}(0,1)$ |
| $I$ | Overall effective external input | 0.3 nA | $\mathcal{U}(0,0.5)$ |
| $G$ | Scaling the strength of network connections | 6.28 | $\mathcal{U}(1,10)$ |
| $\sigma$ | Noise amplitude | 0.005 | $\mathcal{U}(0.0005,0.01)$ |

2024). The pDMF model has also been used to study whole-brain dynamics in various brain disorders, including Alzheimer's disease (**Monteverdi et al., 2023**), schizophrenia (**Klein et al., 2021**), and stroke (**Klein et al., 2021**). The pDMF model equations are given as:

$$\frac{dS_i(t)}{dt} = -\frac{S_i}{\tau_s} + (1 - S_i)\,\gamma\,H(x_i) + \sigma\,\xi_i(t)$$

$$H(x_i) = \frac{ax_i - b}{1 - \exp\left(-d(ax_i - b)\right)} \tag{6}$$

$$x_i = wJS_i + GJ\sum_{j=1}^{N} \mathrm{SC}_{ij}S_j + I$$

where $H(x_i)$, $S_i$, and $x_i$ denote the population firing rate, the average synaptic gating variable, and the total input current at the $i_{th}$ brain region, respectively. $\xi_i(t)$ is uncorrelated standard Gaussian noise and the noise amplitude is controlled by $\sigma$. The nominal parameter values and the prior range for the target parameters are summarized in *Table 6*.

According to recent studies (**Kong et al., 2021**; **Zhang et al., 2024**), we can parameterize the set of $w$, $I$ and $\sigma$ as linear combinations of group-level T1w/T2w myelin maps (**Glasser and Van Essen, 2011**) and the first principal gradient of functional connectivity:

$$w_i = a_w\,\mathbf{Mye}_i + b_w\,\mathbf{Grad}_i + c_w$$

$$I_i = a_I\,\mathbf{Mye}_i + b_I\,\mathbf{Grad}_i + c_I \tag{7}$$

$$\sigma_i = a_\sigma\,\mathbf{Mye}_i + b_\sigma\,\mathbf{Grad}_i + c_\sigma$$

where $\mathbf{Mye}_i$ and $\mathbf{Grad}_i$ are the average values of the T1w/T2w myelin map and the first FC principal gradient, respectively, within the $i_{th}$ brain region. Therefore, the set of unknown parameters to estimate includes $G$ and linear coefficients $(a_w, b_w, c_w, a_I, b_I, c_I, a_\sigma, b_\sigma, c_\sigma) \in \mathbb{R}^9$, hence 10 parameters in total.

### The Balloon-Windkessel model

The Balloon-Windkessel model is a biophysical framework that links neural activity to the BOLD signals detected in fMRI. This is not a neuronal model but rather a representation of neurovascular coupling, describing how neural activity influences hemodynamic responses. The model is characterized by two state variables: venous blood volume ($v$) and deoxyhemoglobin content ($q$). The system's input is blood flow ($f_{in}$), and the output is the BOLD signal ($y$):

$$y(t) \quad = \quad \lambda(\nu, q, E_0) = V_0 \left( k_1(1 - q) + k_2 \left( 1 - \frac{q}{\nu} \right) + k_3(1 - \nu) \right)$$

$$k_1 \quad = \quad 4.3 \, \vartheta_0 E_0 \, \text{TE}$$

$$k_2 \quad = \quad \varepsilon \, r_0 E_0 \, \text{TE} \tag{8}$$

$$k_3 \quad = \quad 1 - \varepsilon$$

where $V_0$ represents the resting blood volume fraction, $E_0$ is the oxygen extraction fraction at rest, $\epsilon$ is the ratio of intra- to extravascular signals, $r_0$ is the slope of the relationship between the intravascular relaxation rate and oxygen saturation, $\vartheta_0$ is the frequency offset at the surface of a fully deoxygenated vessel at 1.5 T, and TE is the echo time. The dynamics of venous blood volume $v$ and deoxyhemoglobin content $q$ are governed by the Balloon model's hemodynamic state equations:

$$\tau_0 \frac{d\nu}{dt} \quad = \quad f(t) - \nu(t)^{1/\alpha}$$

$$\tau_0 \frac{dq}{dt} \quad = \quad f(t) \frac{1 - (1 - E_0)^{1/f}}{E_0} - \nu(t)^{1/\alpha} q(t) \tag{9}$$

where $\tau_0$ is the transit time of blood flow, $\alpha$ reflects the resistance of the venous vessel (stiffness), and $f(t)$ denotes blood inflow at time $t$, given by

$$\frac{df}{dt} = s,$$

where $s$ is an exponentially decaying vasodilatory signal defined by

$$\frac{ds}{dt} = \epsilon x - \frac{s}{\tau_s} - \frac{(f - 1)}{\tau_f} \tag{10}$$

where $\epsilon$ represents the efficacy of neuronal activity $x(t)$ (i.e. integrated synaptic activity) in generating a signal increase, $\tau_s$ is the time constant for signal decay, and $\tau_f$ is the time constant for autoregulatory blood flow feedback (**Friston et al., 2000**). For parameter values, see **Table 7**, taken from **Friston et al., 2000**; **Stephan et al., 2007**; **Stephan et al., 2008**. The resulting time series is downsampled to match the TR value in seconds.

**Table 7.** Parameter descriptions for the Balloon-Windkessel model to map neural activity to the BOLD signals detected in fMRI.

| Parameter | Description | Value |
|---|---|---|
| $\tau_s$ | Rate constant of vasodilatory signal decay in seconds | 1.5 |
| $\tau_f$ | Time of flow-dependent elimination in seconds | 4.5 |
| $\alpha$ | Grubb's vessel stiffness exponent | 0.2 |
| $\tau_0$ | Hemodynamic transit time in seconds | 1.0 |
| $\epsilon$ | Efficacy of synaptic activity to induce signal | 0.1 |
| $r_0$ | Slope of intravascular relaxation rate in Hertz | 25.0 |
| $\vartheta_0$ | Frequency offset at outer surface of magnetized vessels | 40.3 |
| $\varepsilon$ | Ratio of intra- and extravascular BOLD signal at rest | 1.43 |
| $V_0$ | Resting blood volume fraction | 0.02 |
| $E_0$ | Resting oxygen extraction fraction | 0.8 |
| $TE$ | Echo time, 1.5 T scanner | 0.04 |

## Simulation-based inference

In the Bayesian framework (*van de Schoot et al., 2021*), parameter estimation involves quantifying and propagating uncertainty through probability distributions placed on the parameters (prior information before seeing data), which are updated with information provided by the data (likelihood function). The formidable challenge to conducting efficient Bayesian inference is evaluating the likelihood function $p(x \mid \theta)$. This typically involves intractable integrating over all possible trajectories in the latent space: $p(x \mid \theta) = \int p(x, z \mid \theta) dz$, where $p(x, z \mid \theta)$ is the joint probability density of the data $x$ and latent variables $z$, given parameters $\theta$. For whole-brain network models with high-dimensional and nonlinear latent spaces, the computational cost can be prohibitive, making likelihood-based inference with MCMC sampling challenging to converge (*Hashemi et al., 2020*; *Hashemi et al., 2021*; *Jha et al., 2022*).

SBI (*Cranmer et al., 2020*; *Gonçalves et al., 2020*; *Hashemi et al., 2023*), or likelihood-free inference (*Papamakarios et al., 2019*; *Greenberg et al., 2019*; *Brehmer et al., 2020*), addresses issues with explicit likelihood evaluation in complex systems, where it often becomes intractable. The task of density estimation, one of the most fundamental problems in statistics, is to infer an underlying probability distribution based on a set of independently and identically distributed data points drawn from that distribution. Traditional density estimators, such as histograms and kernel density estimators, typically perform well only in low-dimensional settings. Recently, neural network-based approaches have been proposed for conditional density estimation, showing promising results in Bayesian inference problems involving high-dimensional data (*Papamakarios and Murray, 2016*; *Papamakarios and Pavlakou, 2017*; *Greenberg et al., 2019*; *Gonçalves et al., 2020*; *Lueckmann et al., 2021*; *Liu et al., 2021*; *Hashemi et al., 2023*).

Given a prior distribution $p(\vec{\theta})$ placed on the parameters $\vec{\theta}$, $N$ random simulations are generated (with samples from prior), resulting in pairs $\{(\vec{\theta}_i, \vec{x}_i)\}_{i=1}^{N_{sim}}$, where $\vec{\theta}_i \sim p(\vec{\theta})$ and $\vec{x}_i \sim p(\vec{x} \mid \vec{\theta})$ is the simulated data given $\vec{\theta}_i$. By training a deep neural density estimator $F$ (such as NFs; *Papamakarios and Pavlakou, 2017*; *Durkan et al., 2019*), we can approximate the posterior $p(\vec{\theta} \mid \vec{x})$ with $q_{F(\phi, \vec{x})}$ by minimizing the loss function:

$$\mathcal{L}(\phi) = -\sum_{i=1}^{N_{sim}} \log q_{F(\phi, x_i)}(\theta_i) \tag{11}$$

over network parameters $\phi$. Once the parameters of the neural network $F$ are optimized, for observed data $\vec{x}_{obs}$ we can readily estimate the target posterior $q_{F(\phi, \vec{x}_{obs})}(\vec{\theta}) \simeq p(\vec{\theta} \mid \vec{x}_{obs})$. This allows for rapid approximation and sampling from the posterior distribution for any new observed data through a forward pass in the trained network (*Hashemi et al., 2023*; *Hashemi et al., 2024*).

This approach uses a class of generative machine learning models known as NFs (*Rezende and Mohamed, 2015*; *Kobyzev et al., 2019*) to transform a simple base distribution into any complex target through a sequence of invertible mappings. Here, generative modeling is an unsupervised machine learning method for modeling a probability distribution given samples drawn from that distribution. The state-of-the-art NFs, such as MAFs (*Papamakarios and Pavlakou, 2017*) and NSFs (*Durkan et al., 2019*), enable fast and exact density estimation and sampling from high-dimensional distributions. These models learn mappings between input data and probability densities, capturing complex dependencies and multi-modal distributions (*Kobyzev et al., 2019*; *Kobyzev et al., 2020*).

In our work, we integrate the implementation of these models from the open-source SBI tool, leveraging both MAF and NSF architectures. The MAF model comprises five flow transforms, each with two blocks and 50 hidden units, tanh nonlinearity, and batch normalization after each layer. The NSF model consists of five flow transforms, two residual blocks of 50 hidden units each, ReLU nonlinearity, and 10 spline bins. We apply these generative models to virtual brain simulations conducted with random parameters to approximate the full posterior distribution of parameters from low-dimensional data features. Note that we employ a single round of SBI to benefit from amortization strategy rather than using a sequential approach that is designed to achieve a better fit but only for a specific dataset (*Hashemi et al., 2023*; *Hashemi et al., 2024*).

## Sensitivity analysis

Sensitivity analysis is a crucial step for identifying which model parameters influence the model's behavior in response to changes in input (*Hashemi et al., 2018*; *Hashemi et al., 2023*). A local sensitivity can be quantified by computing the curvature of the objective function through the Hessian matrix (*Bates and Watts, 1980*; *Hashemi et al., 2018*). Using SBI, after estimating the posterior for a specific observation, we can perform sensitivity analysis by computing the eigenvalues and corresponding eigenvectors of the following matrix (*Tejero-Cantero et al., 2020*; *Deistler et al., 2021*):

$$M = \mathbb{E}_{p(\theta|x_{obs})}[\nabla_\theta p(\theta|x_{obs})^T \nabla_\theta p(\theta|x_{obs})], \tag{12}$$

which then does an eigendecomposition $M = Q\Lambda Q^{-1}$. A large eigenvalue in the so-called active subspaces (*Constantine, 2015*) indicates that the gradient of the posterior is large in the corresponding direction, suggesting that the system output is sensitive to changes along that eigenvector.

## Evaluation of posterior fit

To assess the reliability of Bayesian inference using synthetic data, we evaluate the posterior z-scores (denoted by $z$) against the posterior shrinkage (denoted by $s$), as defined by *Betancourt et al., 2014*:

$$z = |\frac{\bar{\theta} - \theta^*}{\sigma_{post}}|, \tag{13}$$

$$s = 1 - \frac{\sigma_{post}^2}{\sigma_{prior}^2}, \tag{14}$$

where $\bar{\theta}$ and $\theta^*$ are the posterior mean and the true values, respectively, $\sigma_{prior}$ is the standard deviation of the prior, and $\sigma_{post}$ is the standard deviation of the posterior.

The z-score quantifies how far the posterior mean of a parameter lies from a reference value (e.g. the true value), scaled by the posterior standard deviation. The shrinkage quantifies how much the posterior distribution has contracted relative to the initial prior distribution after learning from data. A small z-score indicates that the posterior estimate is close to the true value, reflecting accurate inference. A large shrinkage value suggests that the posterior is sharply concentrated, indicating that the parameter is well identified. According to these definitions, an ideal Bayesian inference is characterized by z-scores close to zero and posterior shrinkage values close to one, reflecting both accuracy and reliability in the inferred parameters.

## Flexible simulator and model building

A key feature of the VBI pipeline is its modularity and flexibility in integrating various simulators (see *Figure 2*). The *Simulation* module of the VBI pipeline is designed to be easily interchangeable, allowing researchers to replace it with other simulators, such as TVB (*Sanz-Leon et al., 2015*), Neurolib (*Cakan et al., 2023*), Brian (*Stimberg et al., 2019*), and Brainpy (*Wang et al., 2023a*). This adaptability supports a wide range of simulation needs and computational environments, making the VBI a versatile tool for inference in system neuroscience. In particular, the *Simulation* module offers a comprehensive implementation of commonly used whole-brain models. This is a customized version of the implementation from the open-source TVB simulator. While VBI does not encompass all the features of the original TVB, it is mainly designed to leverage the computational power of GPU devices and significantly reduce RAM requirements (see *Figure 2—figure supplement 1*). This optimization ensures that high-performance clusters can be fully utilized, enabling parallel and scalable simulations, as often is required to perform scalable SBI.

## Comprehensive feature extraction

VBI offers a comprehensive toolbox for feature extraction across various datasets. The *Features* module includes but is not limited to: (1) *Statistical features*, including mean (average of elements), variance (spread of the elements around mean), kurtosis (tailedness of the distribution of elements), and skewness (the asymmetry of the distribution of elements), that can be applied to any matrix. (2) *Spectral features*, such as low-dimensional summary statistics of power spectrum density (PSD). (3) *Temporal features*, including zero crossings, area under the curve, average power, and envelope. (4)

*Connectivity features*, including functional connectivity (FC), which represents the statistical dependencies or correlations between activity patterns of different brain regions, and functional connectivity dynamics (FCD), which captures the temporal variations and transitions in these connectivity patterns over time. These calculations are performed for the whole-brain and/or subnetwork (e.g., limbic system, resting state networks). However, since these matrices are still high-dimensional, we use standard dimensional reduction techniques, such as principal component analysis (PCA) on FC/FCD matrices, to extract their associated low-dimensional summary statistics. (5) *Information theory features*, such as mutual information and transfer entropy. Following (*Hashemi et al., 2024*), we use the term *spatio-temporal data features* to refer to both *statistical features* and *temporal features* derived from time series. In contrast, we refer to the *connectivity features* extracted from FC/FCD matrices as *functional data features*. Note that here, 'spatial' does not necessarily refer to the actual spatial characteristics of the data, such as traveling waves in neural fields, but rather to differences across brain regions.

The *Features* module uses parallel multiprocessing to speed up feature calculation. Additionally, it provides flexibility for users to add their own custom feature calculations with minimal effort and expertise, or to adjust the parameters of existing features based on the type of input time series. The feature extraction module is designed to be interchangeable with existing feature extraction libraries such as tsfel (*Barandas et al., 2020*), pyspi (*Cliff et al., 2023*), hctsa (*Fulcher and Jones, 2017*), and scikit-learn (*Pedregosa et al., 2011*). Note that some lightweight libraries such as catch22 (*Lubba et al., 2019*) are directly accessible from the VBI feature extraction module.

## Acknowledgements

This project/research has received funding from the European Union's Horizon Europe Programme under the Specific Grant Agreement No. 101147319 (EBRAINS 2.0 Project) to MH and VJ, No. 101137289 (Virtual Brain Twin Project) to VJ, No. 101057429 (project environMENTAL) to VJ, and government grant managed by the Agence Nationale de la Recherche reference ANR-22-PESN-0012 (France 2030 program) to SP and VJ. We acknowledge the use of Fenix Infrastructure resources, which are partially funded from the European Union's Horizon 2020 research and innovation programme through the ICEI project under the grant agreement No. 800858. The funders had no role in study design, data collection and analysis, decision to publish, or preparation of the manuscript.

## Additional information

### Funding

| Funder | Grant reference number | Author |
| --- | --- | --- |
| European Commission (EBRAINS 2.0 Project) | 10.3030/101147319 | Viktor Jirsa Meysam Hashemi |
| European Commission (Virtual Brain Twin Project) | 10.3030/101137289 | Viktor Jirsa |
| European Commission (EnvironMENTAL) | 10.3030/101057429 | Viktor Jirsa |
| France 2030 program | ANR-22-PESN-0012 | Spase Petkoski Viktor Jirsa |

The funders had no role in study design, data collection and interpretation, or the decision to submit the work for publication.

### Author contributions

Abolfazl Ziaeemehr, Software, Formal analysis, Validation, Investigation, Visualization, Methodology, Writing – original draft, Writing – review and editing; Marmaduke Woodman, Conceptualization, Resources, Data curation, Software, Methodology, Writing – review and editing; Lia Domide, Resources, Data curation, Software, Writing – review and editing; Spase Petkoski, Supervision, Funding acquisition, Methodology, Writing – review and editing; Viktor Jirsa, Conceptualization,

Resources, Supervision, Funding acquisition, Project administration, Writing – review and editing; Meysam Hashemi, Conceptualization, Resources, Software, Supervision, Funding acquisition, Validation, Investigation, Methodology, Writing – original draft, Project administration, Writing – review and editing

## Author ORCIDs
Abolfazl Ziaeemehr ⓘ https://orcid.org/0000-0002-4696-9947
Marmaduke Woodman ⓘ https://orcid.org/0000-0002-8410-4581
Lia Domide ⓘ https://orcid.org/0000-0002-4822-2046
Spase Petkoski ⓘ https://orcid.org/0000-0003-4540-6293
Viktor Jirsa ⓘ https://orcid.org/0000-0002-8251-8860
Meysam Hashemi ⓘ https://orcid.org/0000-0001-5289-9837

Reviewer #1 (Public review): https://doi.org/10.7554/eLife.106194.4.sa1
Reviewer #2 (Public review): https://doi.org/10.7554/eLife.106194.4.sa2
Author response https://doi.org/10.7554/eLife.106194.4.sa3

## Additional files

### Supplementary files
MDAR checklist

### Data availability
No new data were created or analyzed in this study. All code is available on GitHub (https://github.com/ins-amu/vbi copy archived at *Ziaeemehr, 2025*).

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
