## [Editor Report · eLife Assessment]

This paper presents a **valuable** software package, named "Virtual Brain Inference" (VBI), that enables faster and more efficient inference of parameters in dynamical system models of whole-brain activity, grounded in artificial network networks for Bayesian statistical inference. The authors have provided **convincing** evidence, across several case studies, for the utility and validity of the methods using simulated data from several commonly used models, but more thorough benchmarking could be used to demonstrate the practical utility of the toolkit. This work will be of interest to computational neuroscientists interested in modelling large-scale brain dynamics.

---

## [Referee Report · Reviewer #1 (Public review)]

This work provides a new Python toolkit for combining generative modeling of neural dynamics and inversion methods to infer likely model parameters that explain empirical neuroimaging data. The authors provided tests to show the toolkit's broad applicability, accuracy, and robustness; hence, it will be very useful for people interested in using computational approaches to better understand the brain.

Strengths:

The work's primary strength is the tool's integrative nature, which seamlessly combines forward modelling with backward inference. This is important as available tools in the literature can only do one and not the other, which limits their accessibility to neuroscientists with limited computational expertise. Another strength of the paper is the demonstration of how the tool can be applied to a broad range of computational models popularly used in the field to interrogate diverse neuroimaging data, ensuring that the methodology is not optimal to only one model. Moreover, through extensive in-silico testing, the work provided evidence that the tool can accurately infer ground-truth parameters even in the presence of noise, which is important to ensure results from future hypothesis testing are meaningful.

Weaknesses

The paper still lacks appropriate quantitative benchmarking relative to other inference tools, especially with respect to performance accuracy and computational complexity and efficiency. Without this benchmarking, it is difficult to fully comprehend the power of the software or its ability to be extended to contexts beyond large-scale computational brain modelling.

---

## [Referee Report · Reviewer #2 (Public review)]

Summary:

Whole-brain network modeling is a common type of dynamical systems-based method to create individualized models of brain activity incorporating subject-specific structural connectome inferred from diffusion imaging data. This type of model has often been used to infer biophysical parameters of the individual brain that cannot be directly measured using neuroimaging but may be relevant to specific cognitive functions or diseases. Here, Ziaeemehr et al introduce a new toolkit, named "Virtual Brain Inference" (VBI), offering a new computational approach for estimating these parameters using Bayesian inference powered by artificial neural networks. The basic idea is to use simulated data, given known parameters, to train artificial neural networks to solve the inverse problem, namely, to infer the posterior distribution over the parameter space given data-derived features. The authors have demonstrated the utility of the toolkit using simulated data from several commonly used whole-brain network models in case studies.

Strength:

Model inversion is an important problem in whole-brain network modeling. The toolkit presents a significant methodological step up from common practices, with the potential to broadly impact how the community infers model parameters.

Notably, the method allows the estimation of the posterior distribution of parameters instead of a point estimation, which provides information about the uncertainty of the estimation, which is generally lacking in existing methods.

The case studies were able to demonstrate the detection of degeneracy in the parameters, which is important. Degeneracy is quite common in this type of models. If not handled mindfully, they may lead to spurious or stable parameter estimation. Thus, the toolkit can potentially be used to improve feature selection or to simply indicate the uncertainty.

In principle, the posterior distribution can be directly computed given new data without doing any additional simulation, which could improve the efficiency of parameter inference on the artificial neural network is well-trained.

Weaknesses:

The z-scores used to measure prediction error are generally between 1-3, which seems quite large to me. It would give readers a better sense of the utility of the method if comparisons to simpler methods, such as k-nearest neighbor methods, are provided in terms of accuracy.

A lot of simulations are required to train the posterior estimator, which is computationally more expensive than existing approaches. Inferring from Figure S1, at the required order of magnitudes of the number of simulations, the simulation time could range from days to years, depending on the hardware. The payoff is that once the estimator is well-trained, the parameter inversion will be very fast given new data. However, it is not clear to me how often such use cases would be encountered. It would be very helpful if the authors could provide a few more concrete examples of using trained models for hypothesis testing, e.g., in various disease conditions.

---

## [Author Response]

The following is the authors’ response to the previous reviews

**Reviewer #1 (Public review):**
This work provides a new Python toolkit for combining generative modeling of neural dynamics and inversion methods to infer likely model parameters that explain empirical neuroimaging data. The authors provided tests to show the toolkit's broad applicability, accuracy, and robustness; hence, it will be very useful for people interested in using computational approaches to better understand the brain.Strengths:The work's primary strength is the tool's integrative nature, which seamlessly combines forward modelling with backward inference. This is important as available tools in the literature can only do one and not the other, which limits their accessibility to neuroscientists with limited computational expertise. Another strength of the paper is the demonstration of how the tool can be applied to a broad range of computational models popularly used in the field to interrogate diverse neuroimaging data, ensuring that the methodology is not optimal to only one model. Moreover, through extensive in-silico testing, the work provided evidence that the tool can accurately infer ground-truth parameters even in the presence of noise, which is important to ensure results from future hypothesis testing are meaningful.

We appreciate the positive feedback on our open-source tool that delivers rapid forward simulations and flexible Bayesian model inversion for a broad range of whole-brain models, with extensive in-silico validation, including scenarios with dynamical/additive noise.

WeaknessesThe paper still lacks appropriate quantitative benchmarking relative to non-Bayesian-based inference tools, especially with respect to performance accuracy and computational complexity and efficiency. Without this benchmarking, it is difficult to fully comprehend the power of the software or its ability to be extended to contexts beyond large-scale computational brain modelling.

Non-Bayesian inference methods were beyond the scope of this study, as we focused on full posterior estimation to enable uncertainty quantification and detection of degeneracy. Their advantages and disadvantages are briefly discussed in the Introduction and Discussion sections.

**Reviewer #2 (Public review):**
Whole-brain network modeling is a common type of dynamical systems-based method to create individualized models of brain activity incorporating subject-specific structural connectome inferred from diffusion imaging data. This type of model has often been used to infer biophysical parameters of the individual brain that cannot be directly measured using neuroimaging but may be relevant to specific cognitive functions or diseases. Here, Ziaeemehr et al introduce a new toolkit, named "Virtual Brain Inference" (VBI), offering a new computational approach for estimating these parameters using Bayesian inference powered by artificial neural networks. The basic idea is to use simulated data, given known parameters, to train artificial neural networks to solve the inverse problem, namely, to infer the posterior distribution over the parameter space given data-derived features. The authors have demonstrated the utility of the toolkit using simulated data from several commonly used whole-brain network models in case studies.Strength:Model inversion is an important problem in whole-brain network modeling. The toolkit presents a significant methodological step up from common practices, with the potential to broadly impact how the community infers model parameters.Notably, the method allows the estimation of the posterior distribution of parameters instead of a point estimation, which provides information about the uncertainty of the estimation, which is generally lacking in existing methods.The case studies were able to demonstrate the detection of degeneracy in the parameters, which is important. Degeneracy is quite common in this type of models. If not handled mindfully, they may lead to spurious or stable parameter estimation. Thus, the toolkit can potentially be used to improve feature selection or to simply indicate the uncertainty.In principle, the posterior distribution can be directly computed given new data without doing any additional simulation, which could improve the efficiency of parameter inference on the artificial neural network is well-trained.

We thank the reviewer for the careful consideration of important aspects of the VBI tool, such as uncertainty quantification rather than point estimation, degeneracy detection, features selection, parallelization, and amortization strategy.

Weaknesses:The z-scores used to measure prediction error are generally between 1-3, which seems quite large to me. It would give readers a better sense of the utility of the method if comparisons to simpler methods, such as k-nearest neighbor methods, are provided in terms of accuracy. - A lot of simulations are required to train the posterior estimator, which is computationally more expensive than existing approaches. Inferring from Figure S1, at the required order of magnitudes of the number of simulations, the simulation time could range from days to years, depending on the hardware. The payoff is that once the estimator is well-trained, the parameter inversion will be very fast given new data. However, it is not clear to me how often such use cases would be encountered. It would be very helpful if the authors could provide a few more concrete examples of using trained models for hypothesis testing, e.g., in various disease conditions.

We agree with the reviewer that for some parameters the z-score is large, which could be due to the limited number of simulations, the informativeness of the data features, or non-identifiability, and we do address these possible limitations in the Discussion. In line with our previous study, we stick to Bayesian metrics such as posterior z-scores and shrinkage. The application of an amortized strategy needs to be demonstrated in future work, for example in anonymized personalization of virtual brain twins (Baldy et al., 2025).

Ref: Baldy N, Woodman MM, Jirsa VK. Amortizing personalization in virtual brain twins. arXiv preprint arXiv:2506.21155.

**Reviewer #1 (Recommendations for the authors):**
(1) The authors want to keep the term "spatio-temporal" data features to make it consistent with the language they use in their code, even though they only refer to statistical and temporal features of the time series. I stand by my previous comment that this is misleading and should be avoided as much as possible because it doesn't take into account the actual spatial characteristics of the data. At the very least, the authors should recognize this in the text.

We have now recognized this point.

(2) There are still some things that need further clarification and/or explanation:(a) It remains unclear why PCA needs to be applied to the FC/FCD matrices. It was also unclear how many PCs were kept as data features.

We aim to use as many features as possible as a battery of metrics to reduce the number of simulations. The role of each feature can be investigated in future studies. For instance, PCA is used in the LEiDA approach (Cabral et al., 2017) to enhance robustness to high-frequency noise, thereby overcoming a limitation common to all quasi-instantaneous measures of FC. In this work, the default setting was two PCA components.

Ref: Cabral J, Vidaurre D, Marques P, Magalhães R, Silva Moreira P, Miguel Soares J, Deco G, Sousa N, Kringelbach ML. Cognitive performance in healthy older adults relates to spontaneous switching between states of functional connectivity during rest. Scientific reports. 2017 Jul 11;7(1):5135.

(b) It was also unclear which features were used for each model. This is important for reproducibility and to make the users of the software aware of which features are most likely to work best for each model.

We have done our best to indicate the class of features used in each case. This is illustrated more clearly in the notebook examples provided in the repository.

**Reviewer #2 (Recommendations for the authors):**
Thanks for responding to my suggestions. Here is only one remaining point:Section 2.1: Please mention the atlas used to parcellate the brain; without this information, readers won't know what area 88 is in Figure 1, for example.

We have now mentioned this point. In this study we used AAL Atlas.